# Water stable molecular n-doping produces organic electrochemical transistors with high transconductance and record stability

Alexandra F. Paterson[1], Achilleas Savva [1], Shofarul Wustoni [1], Leonidas Tsetseris [2], Bryan D. Paulsen [3], Hendrik Faber[4], Abdul Hamid Emwas[5], Xingxing Chen[4], Georgios Nikiforidis[1], Tania C. Hidalgo [1], Maximillian Moser[6], Iuliana Petruta Maria[6], Jonathan Rivnay [3], Iain McCulloch[4,6], Thomas D. Anthopoulos [4✉] & Sahika Inal [1✉]

From established to emergent technologies, doping plays a crucial role in all semiconducting devices. Doping could, theoretically, be an excellent technique for improving repressively low transconductances in n-type organic electrochemical transistors – critical for advancing logic circuits for bioelectronic and neuromorphic technologies. However, the technical challenge is extreme: n-doped polymers are unstable in electrochemical transistor operating environments, air and water (electrolyte). Here, the first demonstration of doping in electron transporting organic electrochemical transistors is reported. The ammonium salt tetra-n-butylammonium fluoride is simply admixed with the conjugated polymer poly(N,N′-bis(7-glycol)-naphthalene-1,4,5,8-bis(dicarboximide)-co-2,2′-bithiophene-co-N,N′-bis(2-octyldo-decyl)-naphthalene-1,4,5,8-bis(dicarboximide), and found to act as a simultaneous molecular dopant and morphology-additive. The combined effects enhance the n-type transconductance with improved channel capacitance and mobility. Furthermore, operational and shelf-life stability measurements showcase the first example of water-stable n-doping in a polymer. Overall, the results set a precedent for doping/additives to impact organic electrochemical transistors as powerfully as they have in other semiconducting devices.

[1] Organic Bioelectronics Laboratory, Biological and Environmental Science and Engineering Division, King Abdullah University of Science and Technology (KAUST), Thuwal 23955-6900, Saudi Arabia. [2] Department of Physics, National Technical University of Athens, Athens GR-15780, Greece. [3] Department of Biomedical Engineering, Northwestern University, Evanston, IL 60208, USA. [4] King Abdullah University of Science and Technology (KAUST) Solar Center (KSC), Division of Physical Sciences and Engineering, King Abdullah University of Science and Technology (KAUST), Thuwal 23955-6900, Saudi Arabia. [5] Core Labs, King Abdullah University of Science and Technology (KAUST), Thuwal 23955-6900, Saudi Arabia. [6] Department of Chemistry and Centre for Plastic Electronics, Imperial College London South Kensington, London SW7 2AZ, UK. ✉email: thomas.anthopoulos@kaust.edu.sa; sahika.inal@kaust.edu.sa

D oping is a crucial technique for all established and emergent semiconducting devices, used to control charge carrier concentration/conductivity, by purposefully introducing another material (dopant) to add or remove electrons[1–3]. To-date, the only transistor not to explore doping is the electron transporting (n-type) organic electrochemical transistor (OECT). OECTs are made using soft, solution-processable plastics as active semiconductor layers, that support mixed ionic and electronic conduction[4]. This enables OECTs to transduce ionic signals from the water-based, biological world into electronic signals for the technological world with record efficiency (transconductance ($g_m$))[5]. Along with their design flexibility, variety of form-factors and low operating voltages[5], OECTs are receiving great attention for a variety of aqueous-interfacing bioelectronic applications, such as neuromorphic systems, brain implants, in vitro biosignal recorders and biochemical sensors[6–9], as well as electronic switches and amplifiers[5].

However, for OECT-based applications to be commercially viable, electron transporting (n-type) OECTs must be improved significantly[10]. To-date, only three types of n-type organic electronic materials have been successfully implemented in OECTs, with two of these being conjugated polymers[10–13]. The problem arises from the fact that n-type organic semiconductors (OSCs) must have reduction potentials below $\approx -4$ eV[14,15] to be stable in OECT operating environments: air and water (electrolyte). In turn, this makes it extremely difficult for n-type OSCs to undergo efficient, reversible electrochemical oxidation/reduction processes at low voltages ($\leq 1$ V)[10,16], where the latter is critical for avoiding harmful by-products and OECT degradation. These combined difficulties have resulted in n-type OSCs with performance severely lagging behind hole transporting (p-type) OECTs[10,17,18], rendering essential logic circuits and transistor arrays inaccessible.

One particularly poignant characteristic of poor n-type performance is low electron mobility ($\mu$)[17,18]. Given that doping is a well-known technique for increasing/probing intrinsic $\mu$ in transistors[1,2,19–21], it could be—theoretically—an exceptionally useful tool for advancing n-type OECTs, without requiring additional, complex chemical synthesis. However, for n-doping to be a viable device engineering technique, it must improve—as well as retain—n-type $g_m$ in air, aqueous electrolytes and biological fluids. The technical challenge is therefore extreme: first, n-dopants are unstable in air and water, because HOMO/LUMOs above the reduction potential of oxygen/water are required to enable electron transfer/donation from the n-dopant HOMO to host-OSC LUMO[19]. Consequently, current state-of-the-art air-stability for n-doped OSCs is less than a day[22], n-doped conjugated polymers/OSCs have never shown stability in water, and n-doping is rarely used in n-type OSC devices[23]. Second, even successful n-doped OSCs suffer from low doping efficiencies[24], poor dopant dispersion/aggregation and polymer lattice disruption at high dopant concentrations[1,2].

Here, we report state-of-the-art n-type OECT stability, the first water-stable n-doped polymer, the first doped n-type OECT, and the first doped accumulation mode OECT. Simply admixing an n-dopant with a polymer results in synergistic n-doping and morphology-additive effects that improve overall OECT operation. Specifically, we find that n-doping generates delocalized/unpaired electrons, which improve electron mobility and increase ion uptake and storage, whilst the presence of the dopant creates a favourable microstructure, that potentially supports facile ion penetration and migration. Overall, we employ, for the first time in n-type OECTs, a molecular dopant that acts as a simultaneous n-dopant and morphology additive, with the resultant high performance and record stability demonstrating a crucial step in advancing the field of OECTs towards commercialized organic bioelectronic technologies.

## Results

**N-doped organic electrochemical transistors.** Poly($N,N'$-bis(7-glycol)-naphthalene-1,4,5,8-bis(dicarboximide)-$co$-2,2′-bithiophene-$co$-$N,N'$-bis(2-octyldodecyl)-naphthalene-1,4,5,8-bis(dicarboximide) (P-90) (Fig. 1a) was chosen as a host n-type OSC[11,25,26]. This is a promising material for investigating the impact of n-doping in OECTs, because earlier device engineering works suggest there is scope for probing intrinsic P-90 $\mu$, and hence increasing P-90 OECT measured $\mu$[11,27,28]. As an n-dopant, the Lewis basic ammonium salt tetra-n-butylammonium fluoride (TBAF) was chosen (Fig. 1a). TBAF was chosen because it has been shown to n-dope the naphthalene diimide core units in polymers similar to P-90[22,23,29–31], improving $\mu$[31], threshold voltage[23,31] and stability[31] in organic field-effect transistors (OFETs), and improving conductivity in thermoelectrics[22].

We admixed TBAF with P-90 in solution-phase at a range of concentrations (0, 10, 40 and 80 molar percentage (mol%) of the total P-90 molar mass), and tested each system in OECTs gated with NaCl$_{(aq.)}$ electrolyte. Fig. 1b shows the addition of 40 mol% TBAF produces the best-performing OECTs, with a maximum $g_m$ ($g_{m\_max}$) of 10.5 μS, compared to 1.9 μS in 0 mol% OECTs. The performance enhancement is further confirmed in the statistical analysis taken over six OECTs, shown in Supplementary Fig. 1, where the average $g_m$ ($g_{m\_avg}$) is 8.6 μS in 40 mol% TBAF OECTs, compared to 1.7 μS in 0 mol% TBAF OECTs. Supplementary Fig. 2 shows the corresponding transfer and output characteristics of the various transistors, where the reduction in $I_D^{0.5}$ at high $V_G$ (Supplementary Fig. 2b) may occur as a result of $V_G$-dependent contact resistance[2,27]. We note that pristine (i.e. 0 mol% TBAF) $g_{m\_max}$ is higher than previously reported for similar P-90 OECTs[27] and speculate this is because of an effective 1:10 chlorobenzene:chloroform solvent blend (see Experimental Methods)[28]. Taking this into account, P-90:TBAF (40 mol%) $g_{m\_max}$ is $\approx 15 \times$ greater than previously reported for pristine P-90[27] and on par with a previously reported analogue, p(gNDI-gT2), containing an ester group within the glycol side chain, and an alkoxybithiophene unit (gT2) instead of the bithiophene (T2) unit[14,17,18]. The latter is currently the best performing NDI-based OECT material, where Supplementary Table 1 gives an overview of n-type OECT figures of merit reported in the literature. Overall, we report the first use of a dopant for improving transconductance in both n-type and accumulation mode OECTs.

Given that increased $\mu$ is one of the most attractive prospects for doped transistors, and $\mu$ impacts $g_m$[5,32], we started by exploring the effect of TBAF on $\mu$ to explain the increase in $g_m$. Supplementary Fig. 3 shows that a small amount of TBAF (5 mol%) doubles OECT $\mu$, from $8 \times 10^{-5}$ to $16.6 \times 10^{-5}$ cm²/V s. A peak $\mu$ ($18.4 \times 10^{-5}$ cm²/V s) is reached in best-performing P-90:TBAF(40 mol%) devices and decreases at very high concentrations (60–80 mol%). The former rise in $\mu$ could be attributed to trap filling, which would enable the measured $\mu$ values to approach that of the intrinsic P-90 $\mu$. The latter decrease correlates with characteristic OFET doping behaviour, which would typically arise from lattice disturbances introduced by high dopant concentrations[33]. To explore $\mu$ further, we fabricated OFETs comprising the pristine (i.e. P-90 containing 0 mol% TBAF) and best-performing (i.e. P-90 containing 40 mol% TBAF, because 40 mol% results in the highest transconductance OECTs) thin-films. In OFETs, doping-passivated charge traps can increase the number of charge carriers available at low-$V_G$[34–38]. This subsequently increases conductivity, reduces overall resistance[1] and increases $\mu$. Supplementary Fig. 4 shows transfer characteristics of representative P-90:TBAF(0 mol%) and P-90:TBAF(40 mol%) OFETs. Although P-90 does not switch on under traditional field-effects, there is a clear increase in its conductivity with TBAF. The latter phenomenon contributes

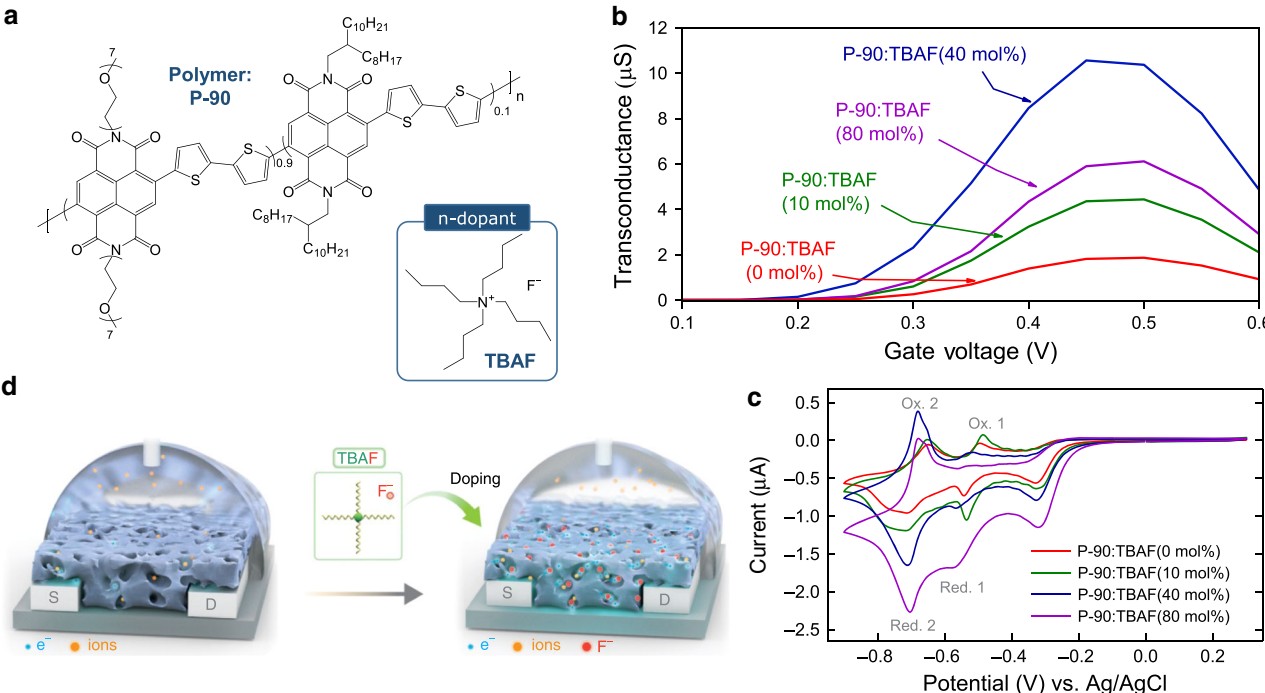

**Fig. 1 Doped electron transporting organic electrochemical transistors. a** Materials used throughout this work: a copolymer made from naphthalene-1,4,5,8-tetracarboxylic diimide (NDI) and bithiophene (T2) with ethylene glycol or alkyl side chains attached to the NDI unit the comonomer, named P-90; the Lewis basic ammonium salt tetra-n-butylammonium fluoride (TBAF) used as an n-dopant. **b** Transconductance of best-performing P-90 organic electrochemical transistors containing 0, 10, 40 and 80 molar percentage (mol%) TBAF. $g_{m\_max}$ is taken at $V_G \approx 0.5$ V and $V_D \approx 0.6$ V, all transistors have a channel length and width of 10 μm and 100 μm, respectively, and channel thicknesses are given in Table 1. **c** Cyclic voltammetry measurements for P-90 thin-films containing 0, 10, 40 and 80 mol% TBAF, measured at a scan rate of 0.05 V/s, with labelled redox peaks. Film thicknesses are given in Table 1. **d** Organic electrochemical transistor configuration and graphical illustration representing TBAF doping mechanisms in P-90, which increase electron density and subsequent ion uptake.

---

**Table 1 A summary of organic electrochemical transistor parameters and electrochemical properties.**

| TBAF (mol%) | Organic electrochemical transistor and electrochemical properties | | | | | | | |
|---|---|---|---|---|---|---|---|---|
| | $g_{m\_max.}$ (μS) | $V_T$ (V) | $I_{ON/OFF}$ | $\mu$ (cm²/V s) | $C^\star$ (F cm⁻³) | $\Delta E_{pp}$ (mV) | $I_{p\_red.}/I_{p\_ox.}$ | Film thickness (nm) |
| 0 | 1.8 | 0.25 | $10^2$ | $8.0 \times 10^{-5}$ | 78 | 77 | 0.87 | 160 |
| 10 | 4.4 | 0.25 | $10^2$ | $1.6 \times 10^{-4}$ | 133 | 75 | 0.91 | 147 |
| 40 | 10.5 | 0.22 | $10^3$ | $1.8 \times 10^{-4}$ | 143 | 28 | 1.00 | 116 |
| 80 | 6.0 | 0.25 | $10^2$ | $1.6 \times 10^{-4}$ | 150 | 25 | 1.01 | 72 |

$g_m$, $V_T$ and $I_{ON/OFF}$ are extracted from $g_m$ vs. $V_G$, $\sqrt{I_{D\_SAT}}$ vs. $V_G$ and $I_{D\_SAT}$ vs. $V_G$, respectively. $\mu$ is extracted using an impedance matching method (see Experimental Section), at $V_G = 0.45$ V and $V_D = 0.6$ V, whilst $C^\star$ is measured using electrochemical impedance spectroscopy (EIS). $\Delta E_{pp}$ and $I_{p\_red.}/I_{p\_ox.}$ are extracted from the cyclic voltammetry (CV) data. The film thicknesses were measured using a mechanical profilometer.

---

to the increased $\mu$ observed in OECTs[38], as well as an increase in $I_{OFF}$ (Supplementary Fig. 1a)[39] and shift in $V_T$ (Supplementary Fig. 2b)[40].

Along with $\mu$, volumetric capacitance ($C^*$) plays a fundamental role in determining $g_m$[5,32]. We therefore used electrochemical impedance spectroscopy (EIS) and cyclic voltammetry (CV) to explore the impact of TBAF on $C^*$. EIS data given in Supplementary Fig. 5 shows that $C^*$ increases with TBAF concentration, with the biggest change occurring between 0 and 10 mol%, and subsequent changes from 10 to 80 mol% occurring gradually. These changes in $C^*$ arise from a combination of the dopant-induced increase in charge carrier density, as well as a reduction in polymer film thickness (see Table 1), where the latter arises from the densification of the polymer upon doping (discussed in detail later) and, in part, from the presence of chlorobenzene used to dissolve the dopant. Figure 1c shows the

corresponding CV curves, highlighting similar features and oxidation/reduction onsets for P-90 containing 0, 10, 40 and 80 mol% TBAF, with the reduction and oxidation peak currents, $I_{p\_red}$ and $I_{p\_ox}$, significantly increasing with TBAF concentration for the second redox couple. The peaks in the CV curves, attributed to electron transfer events along the NDI backbone[14], also indicate ideal peak-to-peak separation ($\Delta E_{pp}$) less than 50 mV and $I_{p\_red.}/I_{p\_ox.}$ ratio = 1 in P-90:TBAF(40 mol%)[41]. This suggests improved reversibility, where electron transfer rate is higher than the rate of mass transport, over the potential window investigated ($-0.8$ to $+0.3$ V vs. Ag/AgCl)[42]. Furthermore, we note that the area enclosed by the CV curve increases with increasing doping concentration; this occurs as the dopant introduces additional charge carriers, which in turn increase capacitance and charge storage properties in the doped systems, in correlation with the EIS data. Table 1 provides an overall

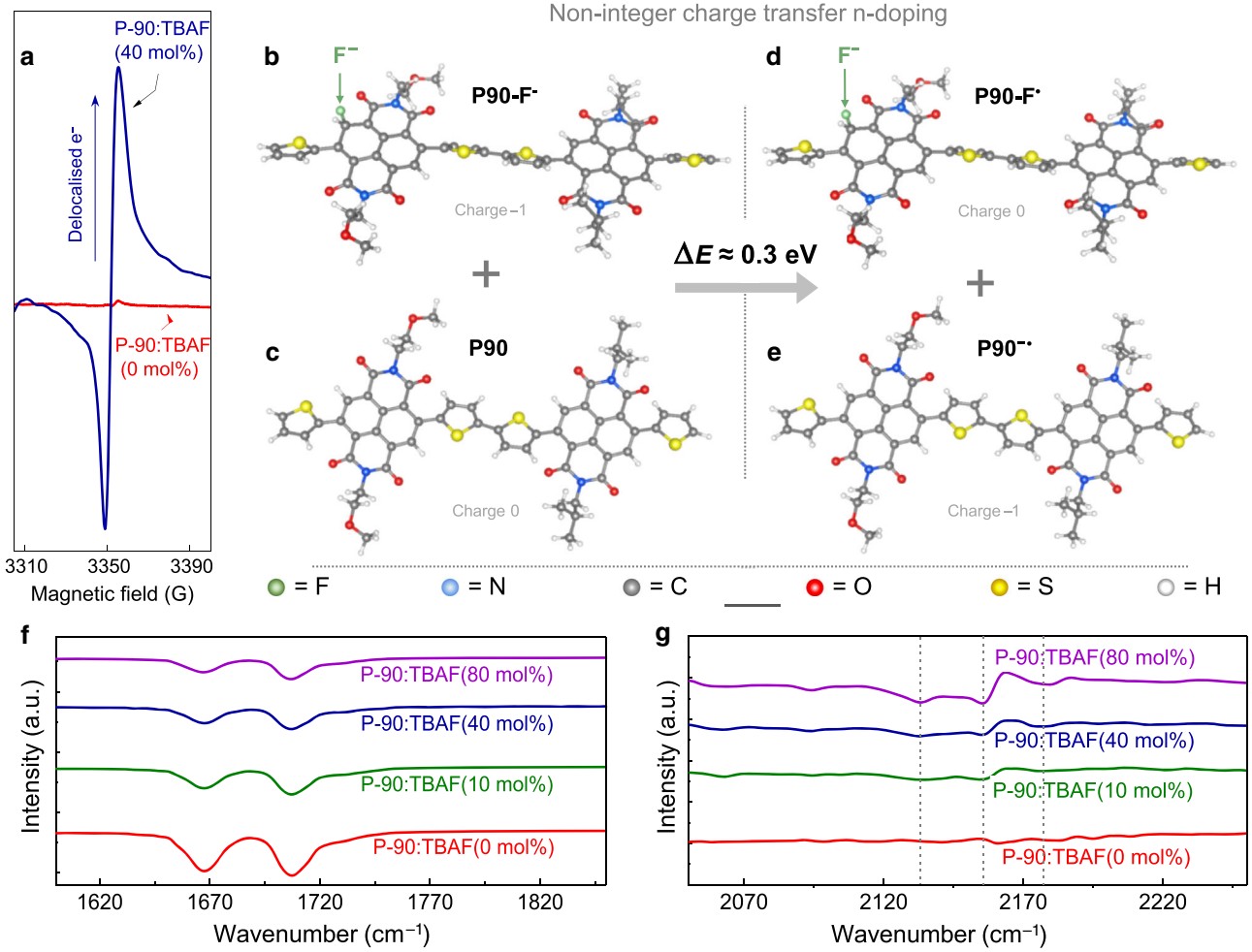

**Fig. 2 TBAF as a molecular n-dopant. a** Electron paramagnetic resonance (EPR) spectroscopy on pristine P-90:TBAF(0 mol%) and best-performing P-90:TBAF(40 mol%). **b–e** TBAF doping mechanisms, calculated using density functional theory: (**b**) a fluorinated P-90 dimer releases an electron to (**c**), an F-free P-90 dimer, by forming a (**d**) fluorinated P-90 radical and (**e**) P-90 anion radical. The energy required for this process to happen ($\Delta E$) is $\approx 0.3$ eV, indicating an electron release could be activated for large concentrations of TBAF. **f–g** Fourier-transform infrared (FTIR) spectroscopy, shifted vertically for clarity: (**f**) symmetric and antisymmetric NDI unit C=O stretching, and (**g**) asymmetrical stretching of –N=C=O in NDI ring unit.

summary of key OECT operating, electrical and electrochemical parameters, showing that TBAF improves OECT characteristics, with P-90:TBAF(40 mol%) being the best-performing system.

**TBAF as a molecular n-dopant.** To verify whether molecular doping is indeed responsible for the improvements in $g_m$, $\mu$ and $C^*$, we used electron paramagnetic resonance (EPR) spectroscopy to observe delocalized electrons by exciting them under a magnetic field, where the delocalization of a paramagnetic electron wave function is typically associated with the broadening of an EPR signal/peak. The results shown in Fig. 2a indicate delocalized electrons are present in best-performing P-90:TBAF(40 mol%) (significant increase in signal intensity), but not in the pristine P-90:TBAF(0 mol%) (flat-line signal)[21,43], suggesting that TBAF indeed acts as an n-dopant in P-90. To elucidate the underlying doping mechanisms, we used calculations within the density functional theory (DFT) approach. The DFT values for the HOMO and LUMO levels indicate that it is not possible for TBAF to transfer an electron to P-90 via integral charge transfer mechanisms, with the $HOMO_{TBAF}$ ($-5.69$ eV) being deeper than $LUMO_{P-90}$ ($-3.79$ eV) (Supplementary Fig. 6). Instead, we find that TBAF can n-dope P-90 through an indirect, two-step mechanism: first, the F$^-$ anion is transferred from TBAF to P-90, with a calculated energy gain $\approx 0.61$ eV, creating a fluorinated

P-90 segment/dimer (P-90-F$^-$) (Fig. 2b). Second, the subsequent P-90-F$^-$ releases an electron to an F$^-$-free P-90 dimer (Fig. 2c). This process results in the formation of a fluorinated P-90 radical and a P-90 anion radical (Fig. 2d, e, respectively) and is described by

$$(\text{P-90} : \text{F}^-)^- + \text{P-90}^0 \rightarrow (\text{P-90} : \text{F}^-)^0 + \text{P-90}^- \qquad (1)$$

The latter P-90 anion radical explains the delocalized electrons observed in P-90:TBAF(40 mol%) with EPR. The former fluorinated P-90 radical is anticipated to undergo other reactions in water, following the work by Weber et al.[44], and therefore would not have an impact on OECT performance. These findings are consistent with previously reported mechanisms from experimental studies on similar n-doped polymers[22,23,31].

Having determined that F$^-$ is responsible for n-doping in P-90 (illustrated in Fig. 1d), we explored the impact of the TBA$^+$ cation on the above mechanisms, finding its location to be critical for successful n-doping. If TBA$^+$ is localized next to the F$^-$ site (Supplementary Fig. 7), the energy required for an electron release is $\Delta E = 2.77$ eV; if TBA$^+$ moves with the electron, $\Delta E$ drops to 1.15 eV (Supplementary Fig. 8), but is still prohibitively high for the n-doping process to be activated. It is only if the TBA$^+$ cation is removed that the energy for electron release drops

drastically to 0.61 eV—a value comparable to the experimental energy required for TBAF to n-dope a fullerene derivative ([6,6]-phenyl-C61-butyric acid methyl ester, PCBM) via similar mechanisms (0.5 eV)[44]. We note that, because the calculated LUMO energy at this level of theory is typically higher than the experimental value (for instance, for the similar N2200 molecule, our DFT LUMO energy is 0.25 eV higher than the experimental value), a more accurate $\Delta E$ estimate is ≈0.30 eV. We therefore conclude that effective doping relies on the TBA$^+$ being withdrawn from the F$^-$ site in P-90. Interestingly, given that the withdrawal of TBA$^+$ is related to the weakening of the Coulomb attraction between the TBA$^+$ cation and the F$^-$-site, it should depend on the solvent and processing conditions once TBAF is admixed into P-90. Further investigations into alternative solvents and processing conditions may therefore optimize doping efficiency and improve overall OECT performance. In the scenario/system presented here, we believe that water soluble TBA$^+$ is removed or reduced to a level below the detection limit during device fabrication and/or operation in the aqueous electrolyte; specifically, when the polymer film is rinsed with deionised water prior to electrochemical measurements.

DFT calculations indicate the HOMO of pristine P-90 (Supplementary Fig. 9a) is significantly less delocalized on the NDI unit, compared to the fluorinated P-90 dimer HOMO (Supplementary Fig. 9b). Likewise, the LUMO of pristine P-90, which receives electrons upon n-doping, has considerable weight on the NDI unit (Supplementary Fig. 9a)—especially compared to the P-90 HOMO. We therefore used Fourier-transform infrared (FTIR) spectroscopy to experimentally probe these differences. The results are summarized in Fig. 2f, g, Supplementary Fig. 10a and Supplementary Table 2, compounding a wealth of evidence to show that delocalized electrons are present in P-90 as a result of adding TBAF:[45] C = O (Fig. 2f) and C = C (Supplementary Fig. 10b) bond vibrations shift/decrease in intensity in TBAF-doped films. The reduced stretching frequency implies conjugation length has been extended, and the double bond profile weakened, as a result of electron delocalization;[46–48] this effect is found to be dependent on TBAF concentration. Furthermore, the vibration of the aromatic ring, sp$^2$ C-H, gradually reduces with the addition of TBAF (Supplementary Table 2, Supplementary Fig. 10c), indicating the presence of delocalized electrons on the NDI-unit[29]. Finally, the vibration band fingerprints at 2154, 2175, 2133 cm$^{-1}$ (Fig. 2g) show that delocalized electrons on the NDI unit form a typical isocyanate group structure (–N = C = O)[49]. We also note that we were unable to detect TBA$^+$ in n-doped P-90, further supporting the withdrawal inferred by DFT, as well as our hypothesis that TBA$^+$ is removed when the polymer film is rinsed with water prior to OECT measurements.

To further support our DFT results on TBAF n-doping mechanisms, as well as demonstrate the broader applicability of this technique, we investigated the impact of TBAF on other NDI-based polymers, namely a poly($N,N'$-bis(7-glycol)-naphthalene-1,4,5,8-bis(dicarboximide)-co-2,2′-bithiophene-co-$N,N'$-bis(2-octyldodecyl)-naphthalene-1,4,5,8-bis(dicarboximide) derivative with extended glycolation (P-100)[11] and the aforementioned NDI homopolymer p(gNDI-gT2)[12]. Using the P-90 best-performing concentration (40 mol%) as a guide, we found that TBAF indeed increases $g_m$ in both polymers. The results are shown in Supplementary Fig. 11. In particular, P-100 $g_{m\_max}$ increases by a remarkable 9 × from P-100:TBAF(0 mol%) to P-100:TBAF(40 mol %), when taking the comparative film thicknesses into account. The thickness-normalized P-100:TBAF(40 mol%) $g_{m\_max}$ is ca. 1.5 S/cm, which is higher than all NDIs reported to-date (see Supplementary Table 1). We also note that $I_{OFF}$ increases and $V_T$ shifts towards more negative voltages, as expected for molecularly n-doped transistors. Although the impact of TBAF is less in

p(gNDI-gT2), we still find that $g_{m\_max}$ doubles, $I_{OFF}$ increases and $V_T$ shifts to more negative voltages with the addition of TBAF (Supplementary Fig. 11d, e); however, substantial OECT device variation inhibits statistical analysis, whilst suggesting that further optimization of the doping concentration is required to harness the intrinsic potential of n-doped p(gNDI-gT2).

Overall, the DFT results suggest that TBAF transfers its F$^-$ to P-90, where the resultant P-90$^-$F$^-$ donates an electron to an F$^-$-free P-90 segment, thereby generating delocalized/unpaired electrons on the P-90 NDI backbone, as experimentally verified with EPR and FTIR. Along with the demonstration that these mechanisms are applicable in other systems, the collective data confirms that this work is the first example of an n-doped OECT, as well as the first n-doped OSC showing stability in water.

**Impact of molecular n-doping on electrochemical doping.** To understand how dopant-generated delocalized/unpaired electrons lead to the enhanced $g_m$ observed in n-type OECTs, we used spectroelectrochemistry, chronoamperometry and electrochemical quartz crystal microbalance dissipation monitoring (EQCM-D) to investigate the impact of TBAF on electrochemical (bulk)-doping (i.e. electron-cation interactions). The following equation characterizes the electrochemical-doping mechanism in the absence of the aforedescribed TBAF chemical doping:

$$\text{P-90}^0 + \text{C}^+ + \text{e}^- \rightleftharpoons \text{P-90}^- : \text{C}^+ \qquad (2)$$

where C$^+$ and e$^-$ represent hydrated cations and electrons, respectively. The electrochemical-doping mechanism in the presence of TBAF-doping can therefore be described by combining Eqs. (1) and (2):

$$\text{P-90}^0 + \text{C}^+ + (\text{P-90} : \text{F}^-)^- + \text{e}^- \rightarrow (\text{P-90} : \text{F}^-)^0 \\ + \text{P-90}^- + \text{P-90}^- : \text{C}^+ \qquad (3)$$

To explore this further, we first used spectroelectrochemistry, to gather real-time information on the interactions between the OSC, electrons and electrolyte cations over a range of electrochemical-doping potentials (−0.2 to 0.6 V). Figure 3a and Supplementary Fig. 12 shows UV–vis absorption spectra changes in pristine P-90:TBAF(0 mol%) compared to best-performing P-90:TBAF(40 mol %) thin-films. During the electrochemical doping of the films up to −0.6 V vs. Ag/AgCl, we observe a slight decrease in the intensity of the region attributed to a $\pi \rightarrow \pi^*$ transition (peak 1, ≈400 nm) and a significant decrease in the intensity of the band attributed to intramolecular charge transfer complex (peak 3, ≈725 nm). While these peaks diminish in intensity, an intermediate peak around 450 nm (peak 2) and a high energy band >800 nm (peak 4) start to appear. We attribute these new features to the polarons formed in the NDI-T2 units[12,50]. The voltage induced changes in the spectral features (Peak 1-4) are consistently greater for the TBAF-dopedP-90 film, indicating more efficient electrochemical doping at all voltages. For instance, a more pronounced decrease in Peak 3 intensity, and increase in Peak 2 and Peak 4 intensity (representative of polaron formation)[50], suggest more efficient coupling between electrolyte cations and the polymer backbone in P-90:TBAF(40 mol%), where the term coupling is used to describe pairing between the ionic and electronic charges[4,51]. We note that the additional changes in these bands, i.e. Peak 2 shifts in wavelength (from ≈ 450 nm to ≈485 nm) and Peak 4 broadens towards higher energy regions in P-90:TBAF(40 mol%), may result from cations interacting with delocalized electrons on the TBAF-doped NDI-unit. Finally, we find that electrochemical doping generates 5.5 mC cm$^{-3}$ in P-90:TBAF(40 mol%), compared to 0.9 mC cm$^{-3}$ in P-90:TBAF(0 mol%) (Fig. 3b), providing further evidence that TBAF enhances electrochemical doping and the charging ability of the n-type film. We note here that doping with

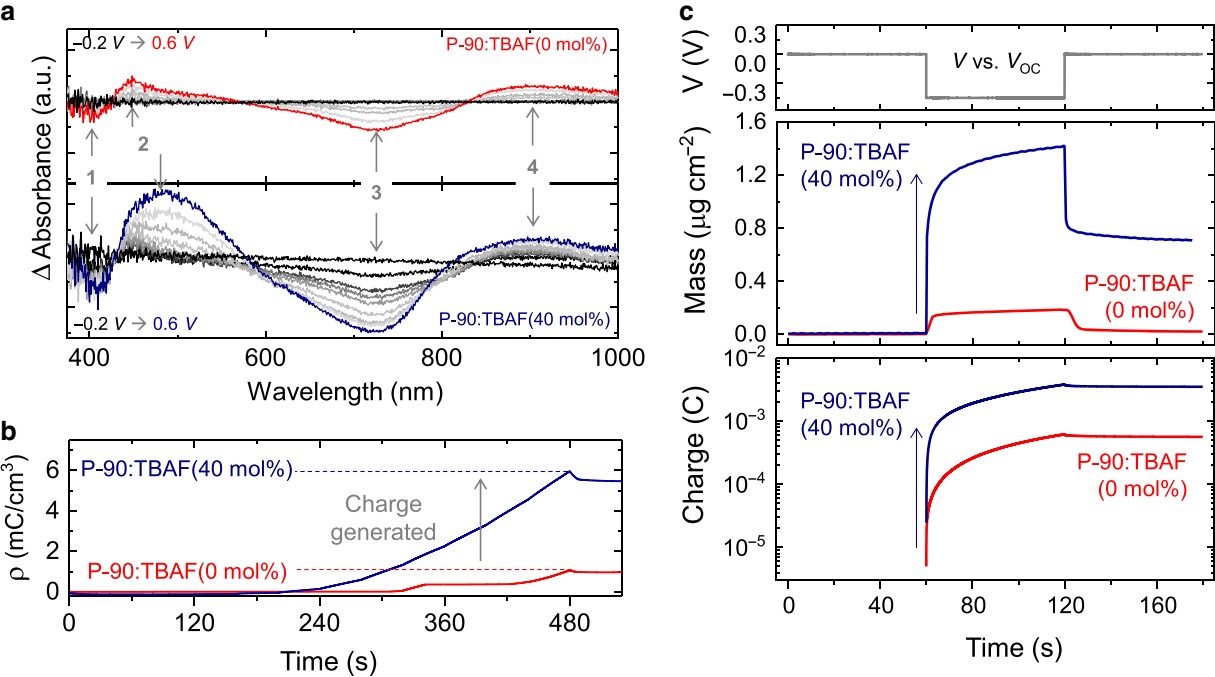

**Fig. 3 Impact of molecular n-doping on electrochemical doping. a** Spectroelectrochemistry measurements showing UV-vis absorption spectra changes with applied voltage (from −0.2 V to 0.6 V in 0.1 V increments) for the pristine P-90 system (0 mol% TBAF), compared to the best-performing, n-doped P-90:TBAF system (40 mol% TBAF). **b** Charge density in 0 mol% TBAF and 40 mol% TBAF thin-films, generated under electrochemical doping potentials applied during spectroelectrochemistry measurements. All measurements are carried out whilst the polymer thin-films are submerged in an aqueous 100 mM NaCl electrolyte solution, and the thin-film thicknesses are given in Table 1. **c** Electrochemical quartz crystal microbalance with dissipation (EQCM-D) monitoring for the pristine P-90 and best-performing P-90:TBAF systems, whilst exposed to 0.1 M NaCl$_{(aq.)}$ electrolyte at an electrochemical-doping potential of 0.45 V vs. $V_{OC}$.

TBAF enables ions to more efficiently couple with electrons inside the polymer film, which can in turn increase OECT $\mu$, as shown in Supplementary Fig. 3.

Next, we used chronoamperometry to calculate ion diffusion coefficients ($D_i$) and EQCM-D to measure water/cation uptake due to electrochemical doping (0.45 V vs. Ag/AgCl)[52]. The former $D_i$ values indicate that ions diffuse ≈ 2 × faster in the TBAF−doped OSC film, with $D_j = 4.873 \times 10^{-10}$ cm s$^{-1}$ in P-90: TBAF(0 mol%) compared to $D_j = 1.894 \times 10^{-9}$ cm s$^{-1}$ in P-90: TBAF(40 mol%); these values also indicate faster electron-cation interactions in P-90:TBAF(40 mol%). For the latter, Fig. 3c and Supplementary Fig. 13 show that, although both systems have an immediate increase in mass upon biasing, as a result of hydrated cations entering the film to compensate for the injected electrons, the P-90:TBAF(40 mol%) mass increases significantly (≈8×) compared to the pristine film (voltage induced mass uptake is 1.42 μg cm$^{-2}$, compared to 0.18 μg cm$^{-2}$). While the films are loaded with hydrated cations, the charge generated corresponds to 0.6 mC and 3.8 mC in P-90:TBAF(0 mol%) and P-90:TBAF(40 mol%), respectively. The greater increase in mass and charge in P-90:TBAF(40 mol%) can be explained by two things: first, an increased number of electrolyte cations drift into TBAF-doped P-90, to compensate for more negative charge (i.e. delocalized/ unpaired electrons). Second, TBAF improves thin-film morphology, allowing water and hydrated cations to drift more easily[52].

To explore the first point, we investigated comparative changes in the polymer thin-films when exposed to NaCl$_{(aq.)}$ electrolyte, with no biasing, using EQCM-D and X-ray photoelectron spectroscopy (XPS). In both cases, we find that P-90:TBAF(40 mol%) draws in Na$^+$ cations from the NaCl$_{(aq.)}$ electrolyte without the need for external biasing to initiate electrochemical doping: EQCM-D (Supplementary Fig. 14) shows P-90:TBAF

(40 mol%) mass increases significantly compared to P-90:TBAF (0 mol%); XPS shows that Na$^+$ is only detected in P−90:TBAF (40 mol%) after overnight submersion in NaCl$_{(aq.)}$ (Supplementary Fig. 15)[53,54], but not in P-90:TBAF(0 mol%). As higher cation content in the active semiconductor can result in increased conductivity, these results also explain the $I_{OFF}$ increase in TBAF-doped OECTs (Supplementary Fig. 2a). Overall, the spectro-electrochemistry, EQCM-D and XPS data indicate significantly more Na$^+$ cations are drawn into the TBAF-doped polymer to compensate for the additional delocalized/unpaired electrons introduced by molecular doping—with and without applying electrochemical-doping potentials. This significantly enhances electrochemical doping, contributing to the increase in $g_m$.

**TBAF as a morphology additive**. The second point implies TBAF may create a preferable P-90 morphology for ion drift[52]. It is therefore important to consider whether TBAF acts as a morphology additive, as well as a molecular dopant. Additives have long been used in OSCs to adapt morphology and control crystallinity, improving molecular ordering and hence electronic properties[55,56]. In OECTs, morphology influences $C^*$, $\mu$ and $g_m$[5,32,57,58], and, in our case, morphology/structural influences are likely because the best-performing TBAF concentrations (40 mol %) are significantly higher than those typically reported in doped organic transistors (≤1 mol%[19–21]). Furthermore, recent works show that OSC microstructure/morphology can be altered by underlying doping mechanisms[3,21,59]. We therefore used atomic force microscopy (AFM) to explore TBAF as an additive in P-90 OECTs, by examining the surface topography of P-90 with varying TBAF concentrations (0, 10, 40 and 80 mol%). The resultant images, shown in Fig. 4a–d and Supplementary Fig. 16,

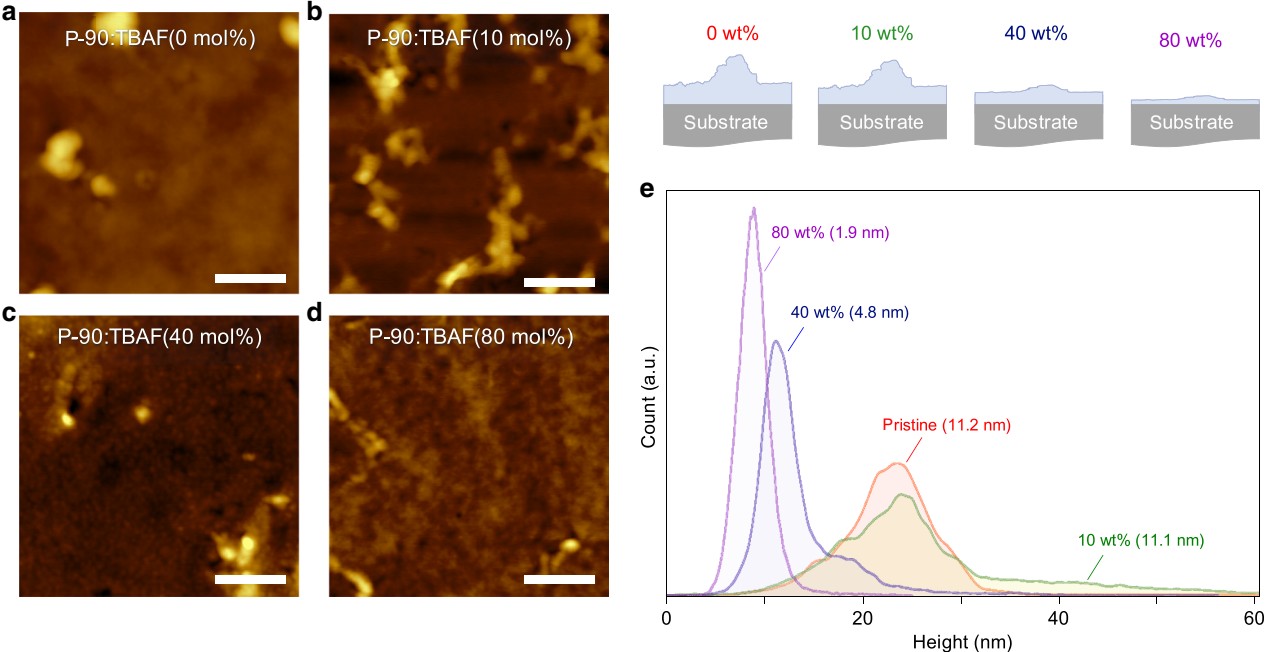

**Fig. 4 The impact of molecular n-doping on morphology.** Atomic force microscopy (AFM) images showing the topography of (**a**) P-90:TBAF(0 mol%), (**b**) P-90:TBAF(10 mol%), (**c**) P-90:TBAF(40 mol%) and (**d**) P-90:TBAF(80 mol%) thin-films. Scale bars are 500 nm. **e** Surface height distribution histograms and surface roughness root mean square (RMS) values corresponding to the topography images in (**a**–**d**). All films were deposited onto 580 × 580 μm² Au electrodes and had been submerged in electrolyte prior to the measurements.

demonstrate that TBAF indeed impacts morphology, by changing surface texture and presence of aggregates. These morphology differences are emphasized even further in the surface height distribution histograms (Fig. 4e), where the pristine layer is seen to contain various peaks, indicating molecular terracing and the presence of large grains. On the other hand, the dopant is seen to reduce and smooth these features. The latter correlates with the surface roughness root mean square (RMS) values, which are found to decrease significantly with increasing TBAF, from 11.2 to 1.9 nm in P-90:TBAF(0 mol%) and P-90:TBAF(80 mol%), respectively (Fig. 4e)[23]. The changes in surface morphology and conversion of large terrains into valleys—or densification of the semiconducting layer—with the addition of TBAF is also expected to impact the polymer film thicknesses (Table 1), eliminate grain boundary effects and potentially facilitate easier ion transport in the film. We expect that this new morphology contributes to the overall device improvements seen. Furthermore, the fact that the TBAF concentration impacts morphology, in addition to carrier concentration, helps elucidate that the optimum doping concentration value of 40 mol% is representative of a synergistic sweet-spot between these two convoluted influences. We note that this value has been found empirically and is expected to be highly dependent on this particular materials system.

Grazing incidence wide angle X-ray scattering (GIWAXS) was used to further explore the impact of TBAF on P-90 structure and morphology. Supplementary Fig. 17 shows both pristine P-90:TBAF(0 mol%) and best-performing P-90:TBAF(40 mol%) have similar in- and out-of-plane scattering; specifically, in- and out-of-plane (100) lamellar scattering, and in-plane backbone (001), (001)′ and (002)′ scattering, in line with previous reports[11,12,28]. The key difference in P-90:TBAF(40 mol%), compared to P-90:TBAF(0 mol%), is a shift in the out-of-plane (100) lamellar peak to lower q, indicating a ~1.5 Å lamellar spacing expansion. The lamellar expansion is small compared to the size of TBAF, suggesting that TBAF intercalation into the crystallites is unlikely, and further supporting the DFT results that only the F⁻ remains in

the polymer film. This may explain the overall lack of disruption to crystallinity, contrary to the significant disruption to polymer ordering which has been observed in highly doped alkylated analogues[60]. On the other hand, the in-plane lamellar and backbone scattering positions and relative intensities show no significant differences. Interestingly, comparing the in- and out-of-plane lamellar expansion, we observe that TBAF doping primarily impacts the edge-on oriented fraction of crystallites. Combined with the AFM images, it appears that the surface roughness change induced by TBAF does not correlate with crystallite structure or orientation. Overall, the GIWAXS results suggest that, while there are no negative effects on crystallinity due to TBAF doping, there are likewise no profound improvements.

**Operational and shelf-life stability.** Finally, to demonstrate molecular n-doping as a commercially viable device engineering technique for OECTs, the TBAF-doped OECTs must operate in biological fluids, withstand sterilization, and display operational and shelf-life stability in both air and water[10]. Practical stability requirements, ranging from a few minutes for disposable sensors to several weeks for cell-culture diagnostic tools[5], are extremely challenging for n-doped OSCs; to-date, the only report of an n-doped OSC exposed to water describes an immediate 5 × decrease in conductivity[61].

We investigated shelf-life and operational stability of sterilized P-90:TBAF(40 mol%) OECTs in phosphate-buffered saline (PBS) —a biologically relevant electrolyte, and a harsher OECT testing environment than pure water. First, we note that, although $g_m$ reduces slightly in PBS compared to NaCl (0.1 M) (from 10.5 to 9.3 μS), and again after 30-min ethanol-sterilization (from 9.3 to 7.2 μS), the state-of-the-art n-type performance is retained (Supplementary Fig. 18a). Next, we measured shelf-life stability, achieving phenomenal results: $g_m$ decreases by only ≈3% after 132 days storage in PBS (Fig. 5a, b and Supplementary Fig. 18b), showcasing state-of-the-art water stability for n-doped OSCs.

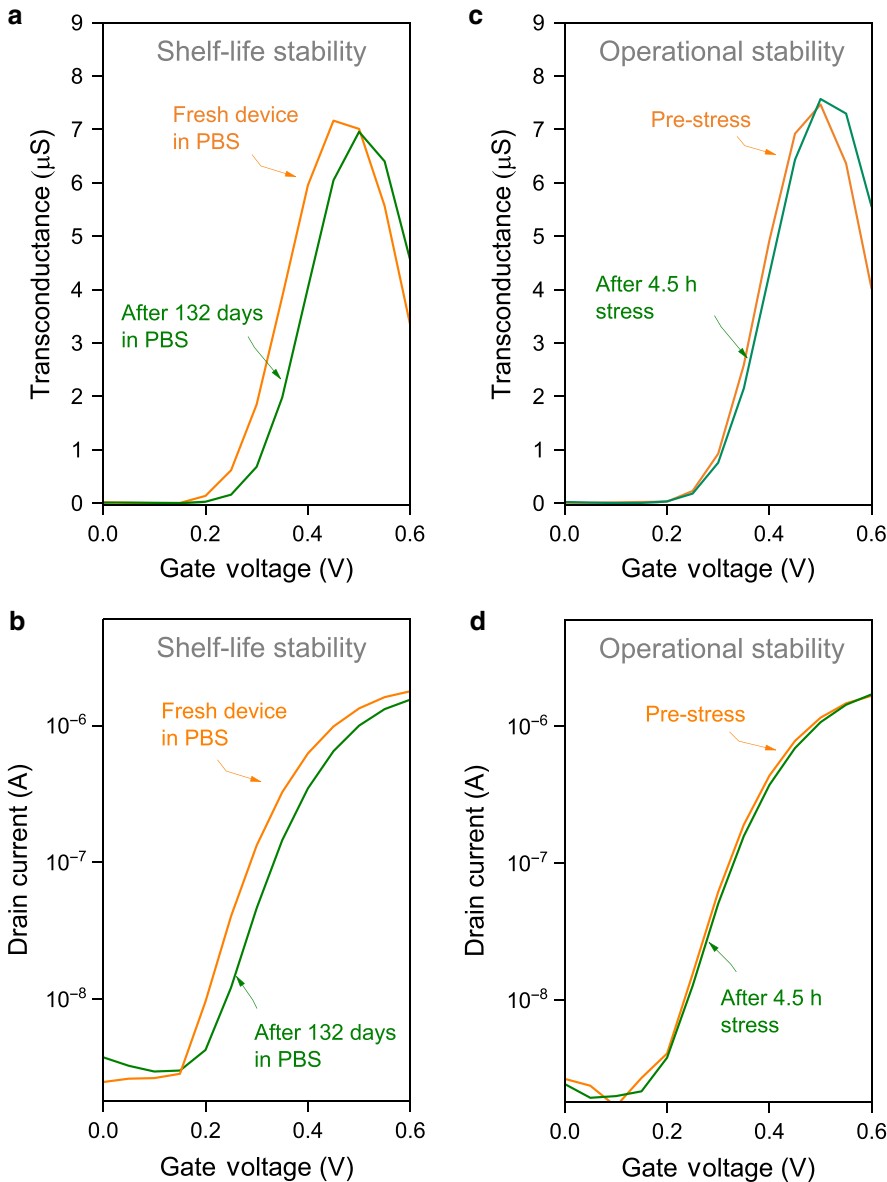

**Fig. 5 Organic electrochemical transistor stability. a** Transconductance and (**b**) transfer characteristics demonstrating the shelf-life stability of best-performing P-90:TBAF(40 mol%) organic electrochemical transistors, after being submerged in phosphate-buffered saline (PBS) solution for 132 days. **c** Transconductance and (**d**) transfer characteristics of P-90:TBAF(40 mol%) organic electrochemical transistors before and after $V_G = 0.4$ V pulsed at 10 s intervals for 4.5 h, demonstrating the operational stability of the best-performing n-doped system, where $V_D = 0.4$ V.

Finally, we demonstrated excellent operational stability; specifically, $g_{m\_max}$ is stable ($\approx 7.5$ µS) after 16,200 s of gate bias pulsed at 10 s intervals ($V_D = V_G = 0.4$ V) (Fig. 5c, d, Supplementary Fig. 19a). We also observe a slight shift in $V_T$, from $\approx 0.25$ V to $\approx 0.26$ V (Supplementary Fig. 18c), that may be associated with electron/ion trapping and/or morphological changes at the electrode/semiconductor interface, as frequently reported for OFETs[62]. However, detailed studies on the latter are generally unavailable for OECTs. Finally, with these experiments we also estimate the response time of the n-doped OECT, via an exponential fit, to be $\approx 24$ ms (Supplementary Fig. 19b). This value, while dependent on channel geometry, is one of the fastest reported so far for n-type OECTs (see Supplementary Table 1).

Overall, the stability experiments show TBAF-doping mechanisms are stable, whilst OECT shelf-life and operational stability fulfill practical requirements[5]. We hypothesize that the stability of the n-doping mechanisms themselves underpin the operational stability; specifically, stable TBAF-doping mechanisms donate charge carriers to fill pre-existing trap states in P-90, which in turn minimize trapping/de-trapping processes during device operation and electrochemical doping, and improve overall operational stability. Furthermore, the combined shelf-life/stability tests are, to the best of our knowledge, the most rigorous reported to date[10,12,63], with the PBS electrolyte presenting a more complex testing environment compared to mono-salt electrolytes. This work therefore reports state-of-the-art shelf life stability in n-type OECTs, as well as state-of-the-art water stability in n-doped OSCs.

## Discussion
In conclusion, we report the first example of molecular doping in n-type OECTs and accumulation mode OECTs. Simply admixing an n-dopant, TBAF, into an n-type conjugated polymer/organic semiconductor, P-90, in solution-phase, is found to produce

n-type OECTs with significantly improved transconductance. The underlying performance improvement is found to arise from synergistic molecular n-doping and morphology effects. In particular, EPR, DFT and FTIR show that $F^-$ from TBAF n-dopes the P-90, generating delocalized/unpaired electrons on the polymer backbone. The impact of the latter is two-fold: firstly, OFET, OECT and CV measurements find the delocalized/unpaired electrons increase mobility, capacitance and conductivity; secondly, spectroelectrochemistry, EQCM-D and XPS show delocalized/unpaired electrons draw more cations from the electrolyte into the organic semiconductor, significantly enhancing electrochemical-doping and electrochemical performance. Two additional polymers were tested with successful improvements, and resulting in the highest transconductance NDI reported to-date. AFM revealed that TBAF smoothens P-90 polymer films, whilst GIWAXS revealed that TBAF does not disrupt molecular ordering, with the former suggesting that TBAF may also act as a morphology additive facilitating ion transport, as well as an n-dopant. Finally, operational and shelf-life stability tests in biological media showcase the first water-stable n-doped organic semiconductor, as well as the most stable n-type OECT to-date (see Supplementary Table 1). The introduction of water-stable n-doping as an easy-to-use technique for enhancing performance in OECTs overcomes many of the existing synthesis challenges acting as a bottleneck to reliable, high-performance electron transporting OECTs and electrolyte/semiconductor interfaces—where such systems are critical if OECTs are to realize their potential in future bioelectronic technologies. Furthermore, the demonstration that water-stable n-doped OECTs have long ambient shelf life and operational stability in biological media is important for their intended use in in vitro and in vivo bioelectronic applications, such as the translation of biological events involving electron exchange (e.g., metabolite sensing)[26], and for the development of biocompatible complementary circuits for on-site amplification of recorded biosignals with low power consumption[10]. The applications of n-type OECTs are manifold but not yet realized due to limited device performance. For instance, n-type OECTs can be ideal platforms to monitor ion channel activity of living cells and lipid bilayers as they specifically respond to cation fluxes. They are also promising to build next generation microscale sensors for cations, which currently rely on decades-old technology[64]. We also note the broader need for easy-to-cast, high mobility, high capacitance and aqueous electrolyte stable n-type mixed conductors in applications beyond bioelectronics, such as organic thermoelectric generators and electrochromic devices as well as electrochemical capacitors, enzyme-based electrocatalytic energy conversion/storage devices, and batteries. Some of these devices integrate p-type counterparts and need stable and high-performance n-type polymer counterparts for pairing to fully benefit from the use of polymers. While it is a synthetic challenge to develop n-type polymers with high stability and capacitance at the electrolyte interface, this work sets the stage for the use of n-doping as a strategy to bring the performance of n-type mixed conductors closer to that of their p-type counterparts. Overall, these results, therefore, set a precedent for molecular doping to advance the field of organic electrochemical transistors into commercialization, as it has for other semiconducting devices.

## Methods

**Spectroscopy**. FTIR spectra were recorded using a Thermo Scientific Nicolet iS10, between 500 and 4000 $cm^{-1}$, at room temperature in attenuated total reflection (ATR) mode. A single spectrum was taken from 64 averaged scans, whilst OMNIC FTIR software was used to correct the baseline. A Bruker EMX PLUS was used to record EPR spectra at 5 K temperatures, 25 dB microwave attenuation, 100 kHz and 5 GHz modulation frequency and amplitude, respectively. XPS was carried out using a Kratos Axis Ultra DLD, with Al Kα X-ray source ($h\nu$ = 1486.6 eV), under $\approx 10^{-9}$ mbar pressure and at 150 W.

**Electrochemical characterization**. An Autolab PGSTAT128N potentiostat-galvanostat was used for CV and EIS, where Ag/AgCl, Pt and a P-90 coated Au electrode were the reference, counter and working electrodes, respectively. CV curves were acquired at a scan rate of 0.05 V s$^{-1}$. EIS was measured from 10 kHz to 0.1 Hz at either $V_{OC}$ or a DC offset potential (doping), with an AC amplitude of 10 mV. EIS spectra were analyzed using MATLAB and Metrohm NOVA software. Effective capacitance and C* were extracted from complex impedance data in accordance with previously reported procedures[28]. E-QCMD was performed using a Q-sense analyzer (QE401, Biolin Scientific) coupled to a potentiostat, whilst spectroelectrochemistry was performed using an Ocean Optics HL-2000-FHSA halogen light source, Ocean Optics QE65 Pro Spectrometer, OceanView software, Keithley 2602 A sourcemeter and a sample holder purchased from redox.me (i.e. MM SPECTRO-EFC). Both were carried out in 0.1 M NaCl$_{(aq.)}$. E-QCMD was measured at 0.5 V vs. $V_{OC}$. Spectroelectrochemistry was measured whilst biasing the films from $-0.2$ V to 0.6 V, in steps of 0.1 V. The charge generated was calculated by measuring I whilst applying V. Ion diffusion coefficients were calculated using the Cottrell equation and following procedures in[52].

**Morphological characterization**. AFM was measured using a Dimension Icon SPM-IAC AFM (Bruker) in tapping mode. 2-D GIWAXS patterns were collected from P-90:TBAF(0 mol%) and P-90:TBAF(40 mol%) thin-films spin-coated onto ~8 mm ×~8 mm Si wafers. Scattering was carried out at the Advanced Photon Source at Argonne National Laboratory on beam line 8-ID-E at room temperature under vacuum with 10.92 keV ($\lambda = 1.135$ Å) synchrotron radiation with a 0.14° incident angle and measured with a Pilatus 1 M hybrid pixel array detector during 5 s exposures. Data analysis was carried out with GIXSGUI Matlab toolbox[65] and customized curve fitting code.

**Density functional theory**. DFT calculations were performed using the code NWCHEM[66], the B3LYP[67,68] exchange-correlation functional and the DZVP basis[69].

**Organic semiconductor solutions**. P-90 was prepared in chloroform at 5 mg/ml. TBAF was prepared in chlorobenzene at 1.8 mg/ml, in an inert nitrogen-filled glovebox. The TBAF solution was added to the P-90 solution, such that TBAF was present in P-90 at varying molar percentages (mol%) of the total P-90 molar mass (12400 g/mol). The admixed P-90:TBAF solutions were left overnight in the glovebox, prior to spin-coating deposition the next day.

**Organic transistors**. Photolithography was used to fabricate Cr (10 nm)/Au (100 nm) source/drain electrodes, microscale electrodes, contact pads and OECT channels with 10 μm and 100 μm channel length and width, respectively. Two layers of Parylene-C were deposited to pattern the OSC in the channel. The source/drain electrodes were treated with UV-ozone for 10 min before spin-coating the OSC solutions at 1000 rpm for 30 s. OECTs were gated using an Ag/AgCl electrode immersed in either 0.1 M NaCl$_{(aq.)}$ or PBS, under identical biasing conditions ($V_D = V_G = 0.6$ V) using a Keithley 2606 A SourceMeter unit. OECT $\mu$ was calculated using impedance matching to measure the transit time of the electronic carriers, and subsequently calculate $\mu$, following procedures outlined in[52]. OFETs were fabricated with evaporated Au source/drain electrodes, at 10 μm and 100 μm channel length and width, respectively, and using Cytop as a dielectric layer. OFETs were biased with $V_D = V_G = 0.6$ V with a Keysight B2912A Precision Source/Measure Unit.

**Shelf-life and operational stability**. OECTs were sterilized by submerging them in an ethanol-filled petri dish, for 30 min, prior to stability tests. Shelf-life stability was recorded by submerging the OECTs in PBS solution and leaving, in a fume hood, whilst carrying out intermittent testing at $V_D = V_G = 0.6$ V, using a Keithley 2606 A SourceMeter, for 132 days. The OECTs were washed with deionised water before and after electrical measurements. Operational stability was measured by applying pulses of $V_D = V_G = 0.4$ V for 16,200 s, at 10 s intervals, using the same equipment.

## Data availability

All data supporting the findings of this study are available within the article and its Supplementary Information, or from the corresponding author upon request.

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

## Acknowledgements

S.I., T.D.A. and I. M. acknowledge King Abdullah University of Science and Technology (KAUST) for their financial support. I.P.M. thanks Alexander Giovannitti for the monomer of p(gNDI-gT2). L.T. acknowledges the use of GRNET high performance computing facility ARIS under project 6055-STEM-2. B.P. and J.R. acknowledge support from the National Science Foundation Grant No. NSF DMR-1751308. The authors would like to thank Joseph Strzalka and Qingteng Zhang for beam line assistance. This research used resources of the Advanced Photon Source, a U.S. Department of Energy (DOE) Office of Science User Facility operated for the DOE Office of Science by Argonne National Laboratory under Contract No. DE-AC02-06CH11357. Fig. 1d was created by Heno Hwang, scientific illustrator at KAUST.

## Author contributions

A.F.P. conceived the project, carried out the experiments, directed the experiments and wrote the manuscript. A.S. performed E-QCMD and chronoamperometry, and contributed to spectroelectrochemistry and stability measurements. S.W. carried out FTIR and XPS measurements. L.T. performed the DFT calculations. B.D.P performed GIWAXS measurements. H.F. and A-H.E. carried out the EPR measurements. X.C. synthesized the P-90. G.N. contributed to CV measurements. T.C.H. fabricated OECTs for polymer coating. J.R. supervised the GIWAXS measurements. I.M. supervised the polymer synthesis. T.D.A. supervised doping experiments, provided dopant materials and supervised the EPR and OFET measurements. S.I. conceived, designed and supervised the project and edited the manuscript. X.C. synthesized the P-90 and P-100 with the help of M.M. and I.P.M. synthesized p(gNDI-gT2). All authors contributed to discussions and manuscript preparation.

## Competing interests

The authors declare no competing interests.
