## [Peer Review File · Nature Communications]

Reviewers' Comments:

Reviewer #1:

Remarks to the Author:

The authors show that chemically doping the active layer of an n-type organic electrochemical transistor (OECT) significantly improves the device performance. Exceptional stability in water and air is also observed, which is in general almost impossible in n-doped semiconductors. A comprehensive set of experiments together with DFT calculations is used to understand the doping mechanism as well as the origin of the enhanced OECT performance. It is concluded that the excess electrons in the doped film draw in more cations (better electrochemical doping), and that the TBAF dopant also improves the morphology for ionic transport. The work is original, topical and complete. It will certainly have a high impact on the OECT community and beyond. Therefore I recommend publication in Nature Communications, if the following minor points are addressed:

- 1)P. 5-6: How thus doping increase the mobility of P-90? I can see that it increases the conductivity by increasing the number of carriers, but what is the mechanism behind the mobility increase? Trap filling? It does not seem to be morphology, since the in-plane crystallinity remains the same (p.14). On a similar line, the authors should explain specifically how the volumetric capacitance is increased by the doping.
- 2)P.8: Maybe specify: How does EPR show that the paramagnetic electrons are delocalized?
- 3)P. 8: The doping mechanism also leads to a fluorinated P-90 radical. Will this have any effect on the OECT device performance?
- 4)P. 9: When TBA⁺ is removed, where does it go during the doping process?
- 5)P. 11: It would be useful to write down the chemical equation for the electrochemical doping mechanism in the presence and absence of the prior chemical doping. This should then be related to the evolution of peaks 1-4 in the bias-induced absorption spectra (Fig. 3a). I don't understand the decrease of polaron and increase of bipolaron band during the electrochemical doping. The explanation of the peak shifts (due to a "new feature/bond") in the doped system should also be better explained.
- 6)P. 11, l. 252: Coupling is not the best term here (implies electronic coupling).

Reviewer #2:

Remarks to the Author:

This manuscript deals with n-type organic electrochemical transistors doped with an ammonium salt. The authors found that increasing the amount of dopant in the n-type semiconductor (P-90), leads to improved device performance.

Despite some novel results, I do not believe this manuscript deserves publication in Nature Communications, for the following reasons.

- 1) Although some trends in transistor performance upon dopant addition are visible, I believe the presented values are not statistically significant. It seems only one transistors per dopant concentration has been measured. The values in table 1 are given without any error bar.
- 2) The effect of the dopant concentration in transistor properties and morphology is not clearly explained. Why the best performance are observed at 40% concentration?
- 3) The AFM micrographs in Figure 4 do not show a clear evidence of morphology effects.
- 4) The cyclic voltammeteries obtained at different dopant concentration show some differences. How are these explained?
- 5) What is the reason why the film thickness decreases upon increasing the dopant concentration.

What is the reason?

6) Figure 1c is not a transfer curve, as indicated in the text.

7) IN page 5, it is unclear the meaning of "pristine and best performing thin films".

8) The property changes are not "gradual" as stated in the text, but mostly occur between 0% and 10% dopant concentration.

Overall, I do not suggest publication of this manuscript in Nature Communications.

Reviewer #3:

Remarks to the Author:

This work has demonstrated the improved performance of n-type OECTs by simply admixing an n-dopant, TBAF, into an n-type conjugated polymer, P-90, which produces a significantly enhanced transconductance compared with the state-of-the-art devices. Various characterization techniques have been used to illuminate the device performance and working mechanism. Recently, research efforts have been intensively focused on improving stability of n-type organic semiconductors for versatile device applications. The topic selected in this work is important, and the manuscript is well organized which makes the paper be understood easily, but both the novelty proposed by this manuscript and mechanism discussion are premature for the publication in Nature Communications. I have some specific comments.

1. The idea that combining the n-type dopant with n-type OSC to optimize the device performance has been extensively reported. Many of them have achieved decent performance in terms of device stability. In addition, the authors claimed that this work is the first doped n-type OECT. The authors should make a very clear comparison between this work and other published n-type OECT, to evidence their statement.

2. Typically, n-type OSC has relatively low LUMO level, causing the fact that electrons are easily captured by water or oxygen in air, and degrading the device stability. The improved shelf-life stability of such n-type device has been demonstrated in this manuscript. Such improvement is very important for practical application, the mechanism of such stability improvement should be illuminated very clearly in the manuscript.

3. In line 196, the author gave a " ΔE drops to 1.29 eV" if the TBA⁺ moves with the electron, but in Figure S7, a ΔE of 1.15 eV was calculated. The authors should check/confirm and explain it.

4. As mentioned in the manuscript, the stability of OECT exposed in air or water is quite challenging but important for practical applications. For this issue, it will be better if the authors can provide stability test (shelf-life and operational stability) of sterilized P-90:TBAF(40 mol%) OECTs in pure water and air. Because according to the discussion about the doping mechanism in the manuscript, ions in the electrolyte also can be drawn into the doped polymer, therefore please verify if the ions from the electrolyte can impact the performance of the OECTs. In addition, to clearly demonstrate the device stability, it would also be necessary to show the variation of drain current as the function of a long time.

5. The on/off current ratio is also important for a transistor device. The authors should present the current ratio from P-90:TBAF with different doping concentrations.

6. In fig.S1(b), a curve bending can be observed at high gate voltage. Such bending usually is observed from the device using high-mobility OSC, otherwise it should be nearly a straight line for the normal OSC. The mobility given by this manuscript is relatively low, why the bending curve can be observed? The authors should explain this.

7. To prove the novelty and importance of this work, it is also very important to show that the technique proposed in this manuscript has a good adaptability with other materials besides the P-90/TBAF system.

REVIEWER #1

REVIEWER: The authors show that chemically doping the active layer of an n-type organic electrochemical transistor (OECT) significantly improves the device performance. Exceptional stability in water and air is also observed, which is in general almost impossible in n-doped semiconductors. A comprehensive set of experiments together with DFT calculations is used to understand the doping mechanism as well as the origin of the enhanced OECT performance. It is concluded that the excess electrons in the doped film draw in more cations (better electrochemical doping), and that the TBAF dopant also improves the morphology for ionic transport. The work is original, topical and complete. It will certainly have a high impact on the OECT community and beyond. Therefore I recommend publication in Nature Communications, if the following minor points are addressed:

AUTHOR'S REPLY: We thank the reviewer for their highly positive comments about our manuscript. Next we will address the specific comments raised.

REVIEWER: 1)P. 5-6: How thus doping increase the mobility of P-90? I can see that it increases the conductivity by increasing the number of carriers, but what is the mechanism behind the mobility increase? Trap filling? It does not seem to be morphology, since the in-plane crystallinity remains the same (p.14).

AUTHOR'S REPLY: We thank the reviewer for suggesting to discuss these two important points and the additional value it adds to the manuscript.

We believe there are two aspects to increasing the P-90 mobility. Firstly, as the reviewer states, the P-90 mobility is increased by the dopant donating charge carriers that fill existing traps in the semiconductor. We see evidence of this by the shift in threshold voltage to more negative values in Figure S1b, as trap-filling enables free charge carriers to become available at lower gate voltages. The trap-filling enables us to further approach the intrinsic mobility of the semiconductor and therefore measure higher charge carrier mobility values.

Secondly, we find that, in the doped system, ions are able to more efficiently electrochemically dope the polymer and couple with electrons, which in turn increases the OECT mobility. This is largely due to the increase in carrier density (which, as the reviewer correctly points out, also impacts the conductivity) drawing in more ions from the electrolyte (increasing its capacitance). The change in surface morphology is also identified to potentially facilitate easier ion transport in the film. We have made changes to the text on page 5 and page 14 to explain that the mobility is increased due additional charge carriers both filling traps and drawing in additional ions.

“The former rise in μ is attributed to trap-filling, supported by the shift in the threshold voltage (V_T) (**Figure S2b**), which enables measured μ values to approach that of the intrinsic P-90 μ .”

“We note here that chemical doping enables ions to more efficiently couple with electrons inside the polymer film, which can in turn increase OECT μ , as shown in Figure S3.”

REVIEWER: On a similar line, the authors should explain specifically how the volumetric capacitance is increased by the doping.

AUTHOR'S REPLY: Again, we thank the reviewer for raising this important point and the additional value it adds to the manuscript. The dopant-induced increase in charge carrier density also increases charge and capacitance; furthermore, we see a reduction in the thickness of the thin-films with increase in doping concentrations (shown in Table 1), which leads to a further increase in the volumetric capacitance, because the reduction in thickness increases the density of the semiconductor layer.

We have made additions on page 6 to emphasise the fact that volumetric capacitance is increased from these combined effects, i.e. increase in charge and reduction in film thickness.

“These changes in C^* arise from a combination of the dopant-induced increase in charge carrier density, as well as a reduction in polymer film thickness (see **Table 1**), where the latter arises from the densification of the polymer upon doping (discussed in detail later) and, in part, from the presence of chlorobenzene used to dissolve the dopant.”

REVIEWER: 2)P.8: Maybe specify: How does EPR show that the paramagnetic electrons are delocalized?

AUTHOR'S REPLY: We thank the reviewer for suggesting to add this additional information to the paper. We have made changes on page 8 to outline that the delocalisation of the paramagnetic electron wavefunction is typically associated with the broadening of the EPR signal peak.

“To verify whether molecular doping is indeed responsible for the improvements in g_m , μ and C^* , we used electron paramagnetic resonance (EPR) spectroscopy to observe delocalized

electrons by exciting them under a magnetic field, where the delocalisation of a paramagnetic electron wave function is typically associated with the broadening of an EPR signal/peak”

REVIEWER: 3)P. 8: The doping mechanism also leads to a fluorinated P-90 radical. Will this have any effect on the OECT device performance?

AUTHOR'S REPLY: We thank the reviewer for raising this important point. Firstly, we will address the nature of the radical stability. The overall scenario that we have followed is the one proposed by Weber et al. in DOI: 10.1039/c3ta14132b, where the radical undergoes rapid chemical reactions in water and the total reaction is exothermic. In the case that the fluorinated P-90 radical undergoes further reactions, we do not expect it to impact OECT performance.

However, we acknowledge that previous studies to date – as well as our own studies – have not identified and/or studied these follow-on chemistry reactions. If the fluorinated P-90 radical remains within the polymer film, it is possible that its presence could impact morphology by alternating the crystallisation properties of the polymer film, and hence the subsequent alignment/structure of the polymer. This, in turn, could indeed impact how ions move through the polymer film; we note that the fluorinated P-90 radical has no charge and therefore is not expected to electrostatically influence ion migration and only influence this transport via a change in morphology. Even in this case we would expect the impact to be minimal or negligible, because the doped polymer film swells significantly ($\approx 59\%$) once exposed to electrolyte, reducing or removing any impact it may have had on the dry-film morphology.

We have made changes to the manuscript on page 9 to explain this without speculating further on the fate of the species and highlight the fact that we have followed the mechanisms proposed by Weber et al.

“...The latter P-90 anion radical explains the delocalized electrons observed in P-90:TBAF(40 mol%) with EPR. Following the work by Weber et al.,⁴⁷ the former fluorinated P-90 radical is anticipated to undergo follow-on reactions in water, and, thus, not have an impact on OECT performance.”

REVIEWER: 4)P. 9: When TBA⁺ is removed, where does it go during the doping process?

AUTHOR'S REPLY: We thank the reviewer for raising this discussion. After exploring this issue using several characterisation techniques (FTIR, Raman and XPS), we were unable to directly probe/detect the TBA⁺ or species from the TBA⁺ in the polymer thin-film. Based on this data, we believe that as a water soluble cation, the TBA⁺ is removed during processing and measurement procedures; specifically, we rinse the polymer film with deionised water prior to electrochemical measurements. Our hypothesis is further supported in the work Schmidt, M. Martina. *Water compatible ionic and polar organic semiconductors for bioelectronics*. 2018. *Universität Bayreuth, PhD dissertation*, where similar systems for electrochemical applications are explored. The authors find that, whilst sodium ions from the electrolyte will diffuse into polymer films, “tetra butyl ammonium ions simultaneously diffuse in the aqueous solution and be hydrated by water molecules”. They confirm this process using energy-dispersive X-ray spectroscopy to identify elements remaining in the polymer film.

We have made changes on page 10 and 11 to state that TBA⁺ is removed – or at least reduced to a level below the detection limit – during device processing/rinsing of the polymer film.

“In the scenario/system presented here, we believe that water soluble TBA⁺ is removed or reduced to a level below the detection limit during device fabrication and/or operation in the aqueous electrolyte; specifically, when the polymer film is rinsed with deionised water prior to electrochemical measurements.”

FTIR results:

“We also note that we were unable to detect TBA⁺ in n-doped P-90, further supporting the withdrawal inferred by DFT, as well as our hypothesis that TBA⁺ is removed when the polymer film is rinsed with water prior to OECT measurements.”

REVIEWER: 5)P. 11: It would be useful to write down the chemical equation for the electrochemical doping mechanism in the presence and absence of the prior chemical doping.

We thank the reviewer for this excellent suggestion.

The electrochemical reactions in a n-type organic semiconductor–aqueous electrolyte system involve the injection of charge carriers (e⁻) and penetration of hydrated cations (C⁺) to compensate for these charges on the backbone of the semiconducting polymer (in this case, P-90). Therefore, in the absence of chemical doping the chemical equation for electrochemical doping is as follows:

As the TBAF dopant is introduced, the semiconducting polymer (P-90) is n-doped through an indirect, two-step mechanism: (i) the F⁻ anion is transferred from TBAF to P-90, creating P-90:F⁻; (ii) P-90:F⁻ releases an electron to an F⁻-free P-90 dimer, forming a fluorinated P-90 radical and a P-90 anion radical. This doping mechanism is described by:

Therefore, the chemical equation for electrochemical doping with C⁺, in the presence of chemical doping, is as follows:

where we assume that, at some point, the fluorinated P-90 radical will undergo follow-on chemistry in water, following the work by Weber et al. in DOI: 10.1039/c3ta14132b.

We've added this information to the manuscript on page 9 and page 13.

REVIEWER: This should then be related to the evolution of peaks 1-4 in the bias-induced absorption spectra (Fig. 3a).

AUTHOR'S REPLY: We thank the reviewer for their suggestion. Earlier studies on NDI polymers have indicated that, as electrolyte cations couple with the polymer backbone upon electrochemical doping, the spectral changes progress from a flat line to the absorption peaks shown in Figure 3a. In *Giovannitti, A., et al. N-type organic electrochemical transistors with stability in water. Nat Commun 7, 13066 (2016) and DOI: 10.1021/acsenergylett.7b01146*, the authors show that these features – and this particular behaviour – occur as the polymer switches from a neutral state to a reduced state, which happens during electrochemical doping.

During electrochemical doping, both the chemically doped sample (P-90:TBAF) and pristine P-90 show the same trends, but with two marked differences: (i) the changes are more intense (larger Δ Absorbance) for the P-90:TBAF(40mol%) and (ii) a red shift in peak 2 (for P-90 at -0.6 V, the maximum of this band is at ca. 445 nm, for P-90:TBAF, it is at ca. 485 nm) as well as a broadening of peak 4 for the P-90:TBAF. While at this point, we are unsure about the origin of these changes, we postulate that they may arise due to the stabilization of the F- doped P-90 units by the cations of the electrolyte. During electrochemical doping, an electron is injected into the neutral P-90 film which is simultaneously compensated by a cation injected from the electrolyte. In the chemically doped P-90, the presence of P-90⁻ may act as new species for the cations to electrostatically interact.

We have amended the description in the text on page 14 to discuss this.

“For instance, a more pronounced decrease in Peak 3 intensity, and increase in Peak 2 and Peak 4 intensity (representative of polaron formation),⁵³ suggest more efficient coupling between electrolyte cations and the polymer backbone in P-90:TBAF(40 mol%), where the term coupling is used to describe pairing between the ionic and electronic charges.^{54,4} We note that the additional changes in these bands, i.e. Peak 2 shifts in wavelength (from \approx 450 nm to \approx 485 nm) and Peak 4 broadens towards higher energy regions in P-90:TBAF(40 mol%), may result from cations interacting with delocalized electrons on the chemically doped NDI-unit.”

REVIEWER: I don't understand the decrease of polaron and increase of bipolaron band during the electrochemical doping.

AUTHOR'S REPLY: We thank the reviewer highlighting that the existing descriptions are unclear. We have now revised the interpretation that the peaks decrease and increase, following existing studies on NDI-based polymers. During the electrochemical doping up to -0.6 V vs. Ag/AgCl, rather than observing a decrease of the polaron and increase of the bipolaron peaks, as the reviewer pointed out, we observe a slight decrease in the pi-pi* transition and a strong decrease in the band attributed to intramolecular charge transfer complex (ICT). While these peaks diminish in intensity, an intermediate peak around 450 nm and a high energy band >800 nm start to appear. We attribute these new features to the polarons formed in the NDI-T2 units. Note that we do not screen the behaviour above -0.6 V, which could show new peaks arising in the high energy region of the ICT band. Trefz et al attributed the peaks at this region to bipolarons (dianions) formed via a 2-step reduction process including the anion (*Trefz, D. et al., J. Phys. Chem. C 2015, 119, 40, 22760-22771*). They suggest that as the voltage hits the potentials corresponding to the second reduction peak (>-1.3 V vs Fc/Fc+ in an organic solvent),

two new peaks appear, one at high energy region and another at ca. 720 nm, while the radical anion (polaron) bands start to decrease. We have now amended the description in the text on page 13-14 to reflect this.

“During the electrochemical doping of the films up to -0.6 V vs. Ag/AgCl, we observe a slight decrease in the intensity of the region attributed to a $\pi \rightarrow \pi^*$ transition (peak 1, ≈ 400 nm) and a significant decrease in the intensity of the band attributed to intramolecular charge transfer complex (peak 3, ≈ 725 nm). While these peaks diminish in intensity, an intermediate peak around 450 nm (peak 2) and a high energy band >800 nm (peak 4) start to appear. We attribute these new features to the polarons formed in the NDI-T2 units^{14,53}.”

REVIEWER: The explanation of the peak shifts (due to a “new feature/bond”) in the doped system should also be better explained.

AUTHOR’S REPLY: We thank the reviewer again for highlighting this. Following our response to the reviewer’s previous comment, we have rewritten our discussion to the possibility that the peak shifts are due to the presence of chemically doped NDI units, rather than a “new feature/bond”.

REVIEWER: 6)P. 11, l. 252: Coupling is not the best term here (implies electronic coupling).

AUTHOR’S REPLY: We thank the reviewer for their constructive feedback and for raising this discussion. We have used the term coupling as it is recognised in the field as the best term for describing the pairing between ionic and electronic charges in mixed conductors (materials that support the conduction of both ions and electrons). For example, *Berggren, M. et al., Ion Electron–Coupled Functionality in Materials and Devices Based on Conjugated Polymers, Adv. Mater. 2019, 31, 1805813* and *Paulsen, B.D. et al. Organic mixed ionic–electronic conductors. Nat. Mater. 19, 13–26 (2020)* outline the use of this term within this context. However, we agree that our manuscript targets a broad audience and therefore we have amended the text on page 14 to avoid any confusion over the use of the term ‘coupling’ and added references to support this point.

“...suggest more efficient coupling between electrolyte cations and the polymer backbone in P-90:TBAF(40 mol%), where the term coupling is used to describe pairing between the ionic and electronic charges.^{54,4}”

REVIEWER #2

REVIEWER: This manuscript deals with n-type organic electrochemical transistors doped with an ammonium salt. The authors found that increasing the amount of dopant in the n-type semiconductor (P-90), leads to improved device performance.

Despite some novel results, I do not believe this manuscript deserves publication in Nature Communications, for the following reasons.

AUTHOR'S REPLY: We thank the reviewer for their consideration of our manuscript. Before we continue with the specific comments raised, we would like to outline the significance surrounding the novel results that the reviewer describes. OECTs have a dominant role in various applications for healthcare-related uses (*Rivnay, J., et al. Organic electrochemical transistors. Nat Rev Mater, 3, 17086 (2018)*). Ever since their discovery in 1985 by Wrighton et al, the range of OECT based applications has grown rapidly, spanning from sensors and actuators, to complementary circuit elements and neuromorphic circuitries. These applications are exciting, but despite the hype they have created, the applications will remain as “proof-of-principle” and the field will eventually deflate if the current limitation in the library of reliable and high-performance materials persists. One critical threat preventing further advances in this field and the maturation of current OECTs is the lack of electron transporting materials.

The first point regarding novelty is the fact that we introduce a new system without the need for complex chemical synthesis. Since n-type polymers were first reported to operate in aqueous electrolyte gated OECTs in 2016 (*Giovannitti, A., et al. N-type organic electrochemical transistors with stability in water. Nat Commun, 7, 13066 (2016)*), there are to-date only three classes of n-type polymers reported to work successfully working in OECTs. The first report is based on an alternating NDI and T2 polymer backbone, similar to the well-known polymer N2200, but functionalized with EG side chains - the P-90 of this work belongs to this class (*Giovannitti, A., et al., The Role of the Side Chain on the Performance of N-type Conjugated Polymers in Aqueous Electrolytes, Chem. Mater. 2018, 30, 9, 2945-2953*). The second n-type OECT material is a ladder-type polymer, poly(benzimidazobenzophenanthroline) (BBL) (*Wang, S., et al. (2016), Thermoelectric Properties of Solution-Processed n-Doped Ladder-Type Conducting Polymers. Adv. Mater., 28: 10764-10771*). We note that, despite having a higher electron mobility than the NDI based materials, BBL is hard to process (insoluble in most solvents) and produces devices with slow switching speeds, that are incompatible with many applications. The third and most recent n-type material reported for OECTs is a small molecule glycolated fullerene (*Bischak, C. G. et al., ACS Appl. Mater. Interfaces 2019, 11, 31, 28138-28144*). The general lack of materials is because it is extremely difficult to synthesize electron transporting materials that operate in water and at the low voltages compatible with biological applications. Here, we offer a new, easy to use system that overcomes these challenges and bypasses the need for complex chemical synthesis, whilst demonstrating doping in accumulation mode OECTs for the first time. During this revision, we also tested the applicability of n-doping for 2 other n-type polymers available to us and now show the generalizability of the n-doping approach.

The second- and third-points regarding novelty are the performance and stability. The limited n-type materials described above exhibit low performance, both in terms of transconductance and stability in the biologically relevant environments, compared to p-type materials. This work is not only the first to show that chemical doping improves the performance of n-type OECTs and produces state-of-the-art OECTs, but it is also the first show an n-type system that has long ambient shelf life and operational stability in biological media. The typical sensitivity of n-type organic conductors to their environment and the instability of n-dopants in general make our findings more significant. No previous reports have focused on long-term device operation in biologically relevant media for n-type polymers. In fact, only very few have performed such studies for their p-type counterparts (mostly with established PEDOT, see *Dijk, G., et al., Stability of PEDOT:PSS-Coated Gold Electrodes in Cell Culture Conditions. Adv. Mater. Technol. 2019, 1900662*). These are therefore very significant studies as they address concerns regarding the durability of such materials if intended for use in vivo or for cutaneous applications.

Finally, the fourth point regarding novelty comes from the broader applications. Although a high-performance n-type material that is stable in electrolyte is very exciting for applications at the biological/electrolyte interface, the applications for this system are not limited to bioelectronics. For example, such materials are required to build electrochemical capacitors, fuel cells, electrochromic devices and batteries that rely on fast and stable charging with electrolyte ions (*Moia, D., et al, Energy Environ. Sci., 2019, 12, 1349-1357; Ohayon, D. et al. Biofuel powered glucose detection in bodily fluids with an n-type conjugated polymer. Nat. Mater. (2019)*). The latter demonstrates that there is a broader need for high performance n-type polymers that operate at the electrolyte/air interface. We note again that our work is the first to show performance improvement for such materials, without relying on a synthetic toolbox.

Next, we will address all of the points that the rejection is based on, showing that each point can be addressed whilst strengthening the manuscript.

REVIEWER: 1) Although some trends in transistor performance upon dopant addition are visible, I believe the presented values are not statistically significant. It seems only one transistors per dopant concentration has been measured. The values in table 1 are given without any error bar.

AUTHOR'S REPLY: We thank the reviewer for their valuable input. We always measure multiple transistors as standard in our experimental procedures (please see *Paterson, A. F., et al., On the Role of Contact Resistance and Electrode Modification in Organic Electrochemical Transistors. Adv. Mater. 2019, 31, 1902291* and *Savva, A., et al., Balancing Ionic and Electronic Conduction for High-Performance Organic Electrochemical Transistors. Adv. Funct. Mater. 2020, 1907657*), and therefore apologise for not including this important information in the original version of our manuscript. We have now included these statistics as a new figure, **Figure S1**, with the corresponding description on page 5.

“The performance enhancement is further confirmed in the statistical analysis taken over 6 OECTs, shown in **Figure S1**, where the average g_m (g_{m_avg}) is 8.6 μS in 40 mol% TBAF OECTs, compared to 1.7 μS in 0 mol% TBAF OECTs.”

REVIEWER: 2) The effect of the dopant concentration in transistor properties and morphology is not clearly explained. Why the best performance are observed at 40% concentration?

AUTHOR’S REPLY: We thank the reviewer for suggesting this point as a discussion.

There are two aspects here that determine the best doping concentration: (i) donated charge carrier concentration and (ii) morphology. In this particular doped system, at 40 mol%, there is a synergistic “sweet-spot” between the donated charge carrier concentration and morphology. Firstly, the doping concentration must be enough so that the correct number of charge carriers fill existing trap states to increase the mobility, and therefore impact OECT performance. If the concentration is not high enough, the trap states will not be sufficiently filled to increase mobility. Secondly, the morphology is influenced by the presence of the dopant, with its particular influence changing depending on the doping concentration.

Overall, for the P-90:TBAF system, we find that 40 mol% is the “sweet-spot” between these two influences, where this specific value has been obtained from empirical observation. It is difficult to experimentally disentangle these two effects and investigate their individual relationship with the doping concentration, as one depends on the other, with 40 mol% is the best balance for their combined effects. For this reasons, other publications typically do not describe the reasoning for exact dopant concentration values and tend to focus on best performing values obtained empirically. The best way to do this is to explore the device performance above and below the “sweet-spot” – as we do in our manuscript – to give a broader picture of how doping impacts electrical properties/device performance (for example: *Panidi, J., et al., Adv. Sci. 2018, 5, 1700290*; *Paterson, A. F., et al., Adv. Electron. Mater. 2018, 4, 1700464*), thereby justifying the reason for the best-performing value. We also note that the specific value of 40 mol% is likely system-based, i.e. it is dependent on variables such as the host material/polymer and the solvent used for solution-processing.

We have made changes to the manuscript on page 17 to reflect the convoluted and empirical nature of the optimum doping concentration, although, based on the reasons outlined here, we do not believe the lack of discussion on why 40 mol% specifically is the best concentration is fair grounds to reject the paper.

“...The changes in surface morphology and conversion of large terrains into valleys – or densification of the semiconducting layer – with the addition of TBAF is also expected to impact the polymer film thicknesses (**Table 1**), eliminate grain boundary effects and potentially facilitate easier ion transport in the film. We expect that this new morphology, combined with trap filling from the n-doping, leads to the overall device improvements seen. Furthermore, the fact that TBAF concentration impacts morphology, in addition to carrier concentration, helps elucidate that the optimum doping concentration value of 40 mol% is representative of a synergistic sweet-spot between these two convoluted influences. We note that this value has been found empirically and is expected to be highly dependent on the particular materials system.”

REVIEWER: 3) The AFM micrographs in Figure 4 do not show a clear evidence of morphology effects.

AUTHOR'S REPLY: We thank the reviewer for highlighting that it would be clearer if we further emphasise the differences shown in the AFM data in Figure 4. These differences in morphology are particularly emphasised in the AFM surface height distribution histogram, originally given in the Supporting Information (Figure S14) and discussed on page 16. Here, we see that the histogram for the pristine layer contains various peaks, indicating molecular terracing and the presence of large grains. On the other hand, the dopant is seen to reduce and smooth these features, which correlates with the systematic decrease in surface roughness (RMS values) with increasing doping concentration. The conversion of “mountainous” terrains to valleys, or densification of the semiconducting layer, eliminates grain boundary effects and, along with trap filling from the n-doping, leads to the overall device improvements seen.

Overall, there are very clear differences in morphology shown in the AFM images. To make this clearer in the manuscript, we have moved the surface height distribution histogram from Figure S14 to the main figure (**Figure 4**), added a schematic showing the changes in surface features and expanded our discussion on page 17/18. We have also done further AFM measurements on the pristine 0 mol% and best-performing 40 mol% doped systems, but over larger scan areas of 10 μm (compared to the original 2 μm scan sizes). These larger images show the differences with the addition of the dopant even further, and correlate with the surface roughness changes and our discussion. The new data is included as a new **Figure S16**.

“The resultant images, shown in **Figure 4a-d** and **Figure S16**, demonstrate that TBAF indeed impacts morphology, changing surface texture and presence of aggregates. These morphology differences are emphasized even further in the surface height distribution histograms (**Figure 4e**), where the pristine layer is seen to contain various peaks, indicating molecular terracing and the presence of large grains. On the other hand, the dopant is seen to reduce and smooth these features. The latter correlates with the surface roughness root mean square (RMS) values, which are found to decrease significantly with increasing TBAF, from 11.2 to 1.9 nm in P-90:TBAF(0 mol%) and P-90:TBAF(80 mol%), respectively (**Figure 4e**)²⁶. The changes in surface morphology and conversion of large terrains into valleys – or densification of the semiconducting layer – with the addition of TBAF is also expected to impact the polymer film thicknesses (**Table 1**), eliminate grain boundary effects and potentially facilitate easier ion transport in the film. We expect that this new morphology, combined with trap filling from the n-doping, leads to the overall device improvements seen. Furthermore, the fact that TBAF concentration impacts morphology, in addition to carrier concentration, helps elucidate that the optimum doping concentration value of 40 mol% is representative of a synergistic sweet-spot between these two convoluted influences. We note that this value has been found empirically and is expected to be highly dependent on the particular materials system.”

REVIEWER: 4) The cyclic voltammeteries obtained at different dopant concentration show some differences. How are these explained?

AUTHOR'S REPLY: We thank the reviewer for asking us to expand on the existing discussion. On page 6 in the manuscript we discuss the cyclic voltammetry curve changes for the films containing various doping concentrations, and characterize these differences by calculating the peak to peak separation and I_{p_red}/I_{p_ox} ratio:

“Figure 1d shows the corresponding CV curves, highlighting similar features and oxidation/reduction onsets for P-90 containing 0, 10, 40, and 80 mol% TBAF, with the reduction and oxidation peak currents, I_{p_red} and I_{p_ox} , significantly increasing with TBAF concentration in the second redox couple. The peaks in the CV curves, attributed to electron transfer along the NDI backbone¹⁴, also indicate ideal peak to-peak separation (ΔE_{pp}) less than 50 mV and I_{p_red}/I_{p_ox} ratio = 1 in P-90:TBAF(40 mol%)⁴¹. This suggests improved reversibility, where electron transfer rate is higher than the rate of mass transport, over the potential window investigated (-0.8 to + 0.3 V vs Ag/AgCl)⁴².”

In addition to this, the area contained within the cyclic voltammetry curve increases as the dopant concentration increases, showing that the system is charged more extensively in the electrolyte. This happens because the dopant introduces charge carriers and improves charge storage – in agreement with the EIS data in Figure S5– and the morphology changes with the addition of the dopant to increase the density of the layer (densification) by improving the smoothness, discussed in our reply to the previous comment. We have now added this information to further expand the above discussion on page 6.

“Furthermore, we note that the area enclosed by the CV curve increases with increasing doping concentration; this occurs as the dopant introduces additional charge carriers, which in turn increase capacitance and charge storage properties in the doped systems, in correlation with EIS data.”

REVIEWER: 5) What is the reason why the film thickness decreases upon increasing the dopant concentration. What is the reason?

AUTHOR'S REPLY: We thank the reviewer for highlighting this interesting point. We believe the change in thickness occurs for two reasons.

Firstly, the solvents used to fabricate the films may have impacted thickness. One of the key points regarding this system is that we simply admix two solutions together prior to spin-coating – one for the host polymer and the other for the dopant – and do not require complex chemical synthesis or fabrication/deposition methods. On page 5 we highlight that the dopant is dissolved in chlorobenzene, whereas the polymer is dissolved in chloroform. Although each solution contains the same quantity of polymer and is the same concentration (5 mg/ml) to be comparable, there is some subtle variation in the quantity of chlorobenzene, in order to add the required amount of dopant for the specific concentrations. (We note that for all of the 0 mol% data, we included chlorobenzene equivalent to the best-performing system, to account for any solvent blend influences on device performance (also discussed on page 5)). Given the higher

boiling point of chlorobenzene (132°C) compared to chloroform (61°C), chlorobenzene takes longer to evaporate during spin-coating and can therefore produce thinner solution-processed films. The latter means films processed from solutions with slightly higher amounts of chlorobenzene, to add the higher quantities of dopant, may be thinner. We note that changes in rheological behaviour of the combined solutions may also be a reason.

The second and main reason for the difference in film thickness with dopant concentration are the changes in microstructure, which lead to densification, as shown by the increase in CV. The stark differences in surface roughness, shown using AFM (Figure 4) also subsequently impact our Dektak measurements, although we measured the film thickness in at least 5 locations and took the average value. We have made changes to the manuscript on page 6 and page 17 to highlight these points and their potential influence on the observed reduction in thickness. However, we also note that – importantly – all results throughout the manuscript have been carefully normalised to take thickness differences into account.

“...These changes in C^* arise from a combination of the dopant-induced increase in charge carrier density, as well as a reduction in polymer film thickness (see **Table 1**), where the latter arises from the densification of the polymer upon doping (discussed in detail later) and, in part, from the presence of chlorobenzene used to dissolve the dopant.”

“The changes in surface morphology and conversion of large terrains into valleys – or densification of the semiconducting layer – with the addition of TBAF is also expected to impact the polymer film thicknesses (**Table 1**), eliminate grain boundary effects and potentially facilitate easier ion transport in the film.”

REVIEWER: 6) Figure 1c is not a transfer curve, as indicated in the text.

AUTHOR'S REPLY: We thank the reviewer for pointing this out and apologise for our typographical error. We have now amended the text on page 5 in the manuscript to say that just Figure S1 shows the collective transfer and output characteristics, not Figure 1c as well. We also added a **new Figure S1**, so the previous figure can now be found as **Figure S2**.

REVIEWER: 7) IN page 5, it is unclear the meaning of "pristine and best performing thin films".

AUTHOR'S REPLY: We thank the reviewer for suggesting that we further clarify the definition of these two terms. We used these terms to describe the system without a dopant/additive (pristine, where the definition of the word pristine is “in its original condition”) and with a dopant/additive at its best-performing concentration of 40 mol% (we refer to as “best-performing”), where the best-performing concentration produces the highest transconductance OECT. We have now made changes to the manuscript on page 5/6 to make sure the definitions of these terms/descriptions are clear for the reader.

“...To explore μ further, we fabricated pristine (i.e. P-90 containing 0 mol% TBAF) and best-performing (i.e. P-90 containing 40 mol% TBAF, because 40 mol% results in the highest transconductance OECTs) thin-films in OFETs...”

REVIEWER: 8) The property changes are not "gradual" as stated in the text, but mostly occur between 0% and 10% dopant concentration.

AUTHOR'S REPLY: We thank the reviewer for highlighting that there is some ambiguity surrounding the use of the word “gradual” for the changes in C^* . We have now amended the text on page 6 to specify that the largest change in C^* occurs between 0 and 10 mol%, and that the following changes from 10 mol%, to 40 mol% and then to 80 mol% occur gradually.

“EIS data given in **Figure S5** shows that C^* increases with TBAF concentration, with the biggest change occurring between 0 and 10 mol%, and subsequent changes from 10 to 80 mol% occurring gradually.”

REVIEWER: Overall, I do not suggest publication of this manuscript in Nature Communications.

AUTHOR'S REPLY: We sincerely hope that a combination of new results, along with the detailed clarifications provided, and the new experiments/additions to the manuscript based on the feedback, will enable the reviewer to reconsider the recommendation. The manuscript reports the first ever use of doping to improve the performance of OECTs and paves the way to a new, novel area of research.

REVIEWER #3

REVIEWER: This work has demonstrated the improved performance of n-type OECTs by simply admixing an n-dopant, TBAF, into an n-type conjugated polymer, P-90, which produces a significantly enhanced transconductance compared with the state-of-the-art devices. Various characterization techniques have been used to illuminate the device performance and working mechanism. Recently, research efforts have been intensively focused on improving stability of n-type organic semiconductors for versatile device applications. The topic selected in this work is important, and the manuscript is well organized which makes the paper be understood easily, but both the novelty proposed by this manuscript and mechanism discussion are premature for the publication in Nature Communications. I have some specific comments.

AUTHOR'S REPLY: We thank the reviewer for their positive comments on our work and the constructive comments. Before we address the individual specific points that have been raised, we would like to explain why we believe that this work is not premature for publication, both in terms of novelty and underlying mechanisms.

The novelty regarding this system is for a number of reasons. Firstly, to-date only three families of n-type OSCs have been shown to work in OECTs, because it is extremely difficult to synthesise electron transporting materials that operate in water and at low voltages. Here, we have bypassed these difficulties using a simple and highly implementable doping technique that does not rely on a synthetic toolbox. This is also the first example of doping in p-type or n-type accumulation mode OECTs, despite the fact that doping has been critical for the commercialisation of other transistors (e.g. MOSFET, TFT). Next, we have demonstrated state-of-the-art performance and stability and that this is the first work to show an n-type system that has long ambient shelf life and operational stability in biological media. No previous reports have focused on long-term device operation in biologically relevant media for n-type polymers and there is limited work available for p-type OECTs. Furthermore, n-doping itself has never been shown to be stable in water, yet we demonstrate with multiple techniques that we have indeed n-doped a water-stable system. Finally, there are a broad range of applications for this system. OECTs have a dominant role in various applications for healthcare-related uses, as well as sensors and actuators, to complementary circuit elements and neuromorphic circuitries. Electron transporting materials, with performance matching more advanced hole transporting materials, are key for these uses, and the significant stability studies address concerns regarding the durability of such materials if intended for use in vivo or for cutaneous applications. Additionally, such high-performance electron transporting materials are required to build electrochemical capacitors, fuel cells, electrochromic devices and batteries that rely on fast and stable charging with electrolyte ions. Given that there are so many novel aspects to this system, we do not believe it is too premature to publish in terms of novelty, and also believe it is highly likely to have significant impact going forward. Furthermore, thanks to the suggestion of the reviewer, we now tested the approach with two other polymers, where we noted the generalizability of the n-doping for other NDI based OECT materials.

In terms of the mechanisms, we have supported the novel results/findings by providing a broad and thorough picture to ensure that the explanation is not premature for publication. We have

done this through a wide range of experimental electrical (OFET, OECT), electrochemical (CV, EIS, spectroelectrochemistry, E-QCM-D) and materials characterisation (FTIR, EPR, XPS, AFM, GIWAXS) techniques. In addition to all these interdisciplinary techniques, we also provide the theory via DFT calculations – where the combination of theory along with experiment is a very new approach for this field. This thorough characterisation enables us to uncover that the mechanism behind the performance improvements can be categorised into three stages: (i) an increase of carrier concentration (where the underlying mechanisms are explained by theoretical analysis) increases mobility, (ii) the increase in carrier concentration improves electrochemical doping efficiency and C^* , (iii) morphology is altered (possibly enhancing ionic transport). By approaching our system from multiple angles, we are able to address an interdisciplinary topic and provide a solid explanation of the underlying mechanisms. Therefore, we do not believe our results are premature to publish.

REVIEWER: 1. The idea that combining the n-type dopant with n-type OSC to optimize the device performance has been extensively reported. Many of them have achieved decent performance in terms of device stability. In addition, the authors claimed that this work is the first doped n-type OECT. The authors should make a very clear comparison between this work and other published n-type OECT, to evidence their statement.

AUTHOR'S REPLY: We thank the reviewer for highlighting this point. We agree that n-dopants have been used to optimise the performance of organic semiconducting electron transporting devices, such as organic field effect transistors, and those resultant devices have achieved decent stability. Where our work varies greatly with the works the reviewer describes, is the fact that we have incorporated the organic semiconductors into electrochemical transistors or OECTs. OECTs have very different fundamental operating mechanisms compared to, for example, the organic thin-film transistor, as ionic charges are involved in the transport mechanisms as well as electronic charges. An even more important difference is the fact that OECTs operate in different environments, namely water and air. In the existing published organic n-type devices, such a presence of water would negatively impact both the n-type polymer and n-dopant, reducing device stability and overall performance. Therefore, although we agree that n-dopants have been extensively reported to improve n-type organic semiconductors device performance as well as stability, this has never been shown successfully in water – let alone been shown in water to produce record performance and stability.

This difficulty of operating n-type devices and n-dopants in water brings us to the discussion about other published n-type OECTs that the reviewer asks for. Because it is extremely difficult to synthesise materials that transport electrons in water, whilst operating at low voltages (under 1 V), to-date, there are only 4 published materials for n-type OECTs: (i) NDI homopolymer (Giovannitti, A. et al. *Nat Commun* **7**, 13066 (2016)), (ii) P-90 (Giovannitti, A. et al. *Chem. Mater.* 2018, **30**, 9, 2945-2953), (iii) fullerene small molecule (Bischak, C. et al. *ACS Appl. Mater. Interfaces* 2019, **11**, 31, 28138-28144) and (iv) BBL (Sun, H. et al., *Adv. Mater.* 2018, **30**, 1704916). This point makes our work even more novel, as we have developed a new, state-of-the-art system that bypasses the need for complex chemical synthesis whilst overcoming the challenges associated with n-doped polymers operating in water. Furthermore, this

technique/approach is simple to implement and therefore presents excellent opportunities to the field.

Therefore, we agree with the reviewers' excellent suggestion to compare our results with existing n-type OECTs in order to emphasise the fact that this is the first work on n-doping in OECTs as well as the novelty. We have now amended the text on page 2 to include a statement highlighting the fact that only three types of materials have been successfully integrated into OECTs to-date, and only two of them being conjugated polymers.

“To-date, only three types of electron deficient n-type organic electronic materials have been successfully implemented in OECTs, with two of these being conjugated polymers^{12,13,14,15}. The problem arises from the fact that n-type organic semiconductors (OSCs) must have reduction potentials below ≈ -4 eV^{16,17} to be stable in OECT operating environments: air and water (electrolyte). In turn, this makes it extremely difficult for n-type OSCs to undergo efficient, reversible electrochemical oxidation/reduction processes at low voltages (≤ 1 V)^{12,18}, where the latter is critical for avoiding harmful by-products and OECT degradation.”

REVIEWER: 2. Typically, n-type OSC has relatively low LUMO level, causing the fact that electrons are easily captured by water or oxygen in air, and degrading the device stability. The improved shelf-life stability of such n-type device has been demonstrated in this manuscript. Such improvement is very important for practical application, the mechanism of such stability improvement should be illuminated very clearly in the manuscript.

AUTHOR'S REPLY: We thank the reviewer for highlighting the importance of the record n-type OECT stability. Indeed, this is one of the most exciting aspects of our work/results.

Firstly, we will clarify that, in other n-type OSC devices and organic thin-film transistors, stability is reduced in the presence of water because the n-type OSC has a low lying LUMO and electrons can be trapped by water and air, as stated by the reviewer. In our case, we are using devices that utilise different operating mechanisms and therefore different mechanisms influence their stability, as compared to n-type organic thin-film transistors (OTFTs). The OSCs for OECTs are designed to operate in aqueous electrolyte, and here we are using an NDI core polymer with a high density of glycol side chains, with the purpose of the latter to facilitate ion injection from the electrolyte. These ions electrochemically (de)dope the polymer, generating (or compensating for) electrons in the channel, enabling the channel to become conductive and therefore switching the device on. This process that we describe determines device operation and also impacts device stability. Such differences between OECTs and OTFTs are, to-date, largely unexplored and indeed very interesting, with key things such as differences in stability and mobility not yet understood.

One very interesting aspect of this work regarding stability, is the stability of the actual n-doping processes themselves. As the reviewer states, in OTFT applications, the key water-related problem is the low lying LUMO level of n-type OSCs. This problem is even further pronounced for n-dopants that support traditional integer charge transfer, as they require a lower HOMO level than the LUMO of the host OSC, in order to donate electrons into the host LUMO. This

makes their LUMO even lower than the materials the reviewer describes, and they are therefore even more unstable from oxygen and water capturing electrons. Here, we have overcome this bottleneck by using a material that dopes via an “indirect” two-step mechanism rather than direct integer charge transfer, as shown in our DFT calculations. This indirect process is similar to that described for Lewis acids in p-type OTFTs, but in our case n-type we have employed a Lewis base. We believe that it is likely this stable n-doping mechanism in turn impacts OECT stability. Specifically: additional charge carriers introduced by the dopant fill pre-existing trap states in the OSC, as shown by the shift in the threshold voltage. This minimises trapping/de-trapping processes occurring during device operation, and subsequently improves operational stability.

We agree with the reviewer that this is very important, especially considering that other n-type device stability decreases when water is present, and we have now included a discussion on page 20 about these mechanisms and their relationship to the improved n-doped OECT stability. We do note here however that in general OECT stability mechanisms are largely unexplored, and is indeed an interesting area of our ongoing research.

“Overall, the stability experiments show TBAF-doping/additive mechanisms are resistant to degradation, whilst the n-doped OECT shelf-life and operational stability fulfill practical requirements⁵. We hypothesize that the stability of the n-doping mechanisms themselves underpin the operational stability; specifically, stable TBAF-doping mechanisms donate charge carriers to fill pre-existing trap states in P-90, which in turn minimise trapping/de-trapping processes during device operation and electrochemical doping, and improve overall operational stability. Furthermore, the combined shelf-life/stability tests are, to the best of our knowledge, the most rigorous reported to-date^{12,14,67}, with the PBS electrolyte presenting a more complex testing environment compared to mono-salt electrolytes.”

REVIEWER: 3. In line 196, the author gave a “ ΔE drops to 1.29 eV” if the TBA⁺ moves with the electron, but in Figure S7, a ΔE of 1.15 eV was calculated. The authors should check/confirm and explain it.

AUTHOR’S REPLY: We thank the reviewer for highlighting this and apologise for the typographical error. The correct value is indeed 1.15 eV and we have now amended the text to reflect this.

REVIEWER: 4. As mentioned in the manuscript, the stability of OECT exposed in air or water is quite challenging but important for practical applications. For this issue, it will be better if the authors can provide stability test (shelf-life and operational stability) of sterilized P-90:TBAF(40 mol%) OECTs in pure water and air. Because according to the discussion about the doping mechanism in the manuscript, ions in the electrolyte also can be drawn into the doped polymer, therefore please verify if the ions from the electrolyte can impact the performance of the OECTs.

AUTHOR’S REPLY: We thank the reviewer for considering our manuscript and providing valuable comments, although we are confused by the request to verify that ions impact the

performance of the OECT. Ions from the electrolyte are well reported to fundamentally determine OECT operation, and such mechanisms have been covered in detail in a broad range of literature publications and review articles (see DOI:10.1038/natrevmats.2017.86). In our original manuscript, we already show shelf-life and operational stability of sterilized P-90:TBAF(40 mol%) OECTs in water and air, and our stability measurements are purposefully done in an even harsher testing environment than pure water because we use phosphate-buffered saline (PBS). PBS is a water-based salt solution containing disodium hydrogen phosphate, sodium chloride, potassium chloride and potassium dihydrogen phosphate. All OECT measurements in our work are carried out in air. Indeed, throughout the manuscript, the device studies alongside all the electrochemical investigations (EQCM-D, CV, EIS, spectroelectrochemistry) have been performed in both aqueous electrolytes and air.

We have made amendments to the text on page 19/20 to highlight that the extremities of the stability testing-conditions, some of these are found as in the following:

“...We investigated shelf-life and operational stability of sterilized P-90:TBAF(40 mol%) OECTs in phosphate-buffered saline (PBS) – a biologically relevant electrolyte, as well as a harsher OECT testing environment than pure water.”

“Furthermore, the combined shelf-life/stability tests are, to the best of our knowledge, the most rigorous reported to-date^{12,14,67}, with the PBS electrolyte presenting a more complex testing environment compared to mono-salt electrolytes.”

REVIEWER: In addition, to clearly demonstrate the device stability, it would also be necessary to show the variation of drain current as the function of a long time.

AUTHOR’S REPLY: We thank the reviewer for their excellent suggestion and have now included the drain current as a function of time in a new figure, **Figure S19**, corresponding to the data in Figure 5. In regards to the “long time”, we note here that we chose a 4.5 hour cut-off to ensure the volume of the electrolyte remains constant throughout our experimentation by avoiding evaporation. We monitored the latter situation carefully, as any changes in volume will impact the OECT characteristics and therefore our results. We also note that, unlike OTFTs, OECTs do not target applications that require very long-term operational stability and tend to target applications where shelf-life stability is more critical.

“Finally, we demonstrated excellent operational stability; specifically, g_{m_max} is stable ($\approx 7.5 \mu S$) after 16,200 seconds of gate bias pulsed at 10 second intervals ($V_D = V_G = 0.4 V$) (**Figure 5c-d, Figure S19**).”

REVIEWER: 5. The on/off current ratio is also important for a transistor device. The authors should present the current ratio from P-90:TBAF with different doping concentrations.

AUTHOR’S REPLY: We thank the reviewer for suggesting to include a summary of the impact of doping on the on-off current ratio of the OECTs. We have now included this information in revised **Table 1**.

REVIEWER: 6. In fig.S1(b), a curve bending can be observed at high gate voltage. Such bending usually is observed from the device using high-mobility OSC, otherwise it should be nearly a straight line for the normal OSC. The mobility given by this manuscript is relatively low, why the bending curve can be observed? The authors should explain this.

AUTHOR'S REPLY: We thank the reviewer for raising this discussion. Although the reviewer describes something that is valid for OTFTs, this may not necessarily be the case for OECTs. Specifically, in OTFTs, the “curve bending” described happens because the potential barrier at the metal/semiconductor interface (contact resistance) varies with gate voltage. This causes an increase in drain current at lower gate voltages, as the contact resistance decreases at low V_G and therefore the overall transport in the transistor is dictated by how charges are injected at the contact. It has been shown that, when this effect is superimposed onto charge accumulation in the channel, it results in a superlinear increase in I_D at low gate voltages. Eventually V_G is increased enough and the height of the Schottky barrier is reduced such that carriers can be injected as they would with an Ohmic contact. In this higher V_G region, we subsequently observe a decrease in I_D . The overall V_G -dependent increase in I_D followed by a decrease in I_D is described in this comment as “curve bending”. In OTFTs, the nature and extremity of the “curve bending” behavior is determined by the size of the voltage-dependent contact resistance in relation to the channel resistance. “Curve bending” is particularly pronounced when the channel resistance is low and contact resistance decreases sharply to the channel resistance. The latter is more relevant in high mobility OSCs because the channel resistance is lower than in “normal” or low mobility OSCs. This is why the reviewer describes “curve bending” as a feature of high mobility OSCs, but in fact it is dependent on the device itself, and the fact that we have used a low mobility OSC does not necessarily mean that there should be a linear relationship between $\sqrt{I_D}$ and V_G . These effects in detail in *Paterson, A. F., et al., Adv. Mater. 2018, 30, 1801079*.

Similar to stability, there are nuances and differences between contact resistance in OECTs and contact resistance in OTFTs. To-date there are only three investigations into the effects of contact resistance on OECT performance (*Kaphle, V., et al. (2016), Contact Resistance Effects in Highly Doped Organic Electrochemical Transistors. Adv. Mater., 28: 8766-8770; Paterson, A. F., et al., On the Role of Contact Resistance and Electrode Modification in Organic Electrochemical Transistors. Adv. Mater. 2019, 31, 1902291; Friedlein, J. T., et al., Influence of disorder on transfer characteristics of organic electrochemical transistors Appl. Phys. Lett. 111, 023301 (2017)*). In one of these publications, we explored the impact of contact resistance in n-type OECTs and found that there are two key contact resistance regions: at lower V_G when contact resistance can influence device performance, and at higher V_G when the ions have been drawn into the films from the electrolyte to electrochemically dope the OSC, subsequently reducing the width of the Schottky barrier at the metal/semiconductor interface and enabling Ohmic-like charge injection. Following this, it is therefore possible that we could observe a similar “curve bending” or reduction in drain current at higher gate voltages in OECTs, as in OTFTs, based on the mechanisms we have described above. We have therefore added details into the text on page 5 regarding the changes in $\sqrt{I_D}$ with V_G .

REVIEWER: 7. To prove the novelty and importance of this work, it is also very important to show that the technique proposed in this manuscript has a good adaptability with other materials besides the P-90/TBAF system.

AUTHOR'S REPLY: We thank the reviewer for another excellent suggestion. However, we stress here that, unlike OTFTs, there are a very limited number of available materials for n-type OECTs. Despite this, we have now tested the TBAF in two more systems, showcasing the apparent universality of our approach. Using the optimum/best-performing 40 mol% concentration in P-90 as a guide, the results for P-100 (another P-90 derivative, *Giovannitti, A. et al. Chem. Mater. 2018, 30, 9, 2945-2953*) and p(gNDI-gT2) (an NDI homopolymer, *Giovannitti, A. et al. Nat Commun 7, 13066 (2016)*) are shown in **new Figure S11** for 0 and 40 mol%, and discussed on page 11/12. We found that, in both cases, the addition of the TBAF increases the transconductance, with a remarkable increase of 9x in the P-100. Although we find some limitations on the targeted materials, in particular p(gNDI-gT2), we envision the use of different dopants for different channel materials, and by no means restrict our ongoing choices to our model system of P-90 and TBAF.

“To demonstrate the broader applicability of this technique, we investigated the impact of TBAF on other NDI-based polymers, namely a poly(N,N'-bis(7-glycol)-naphthalene-1,4,5,8-bis(dicarboximide)-co-2,2'-bithiophene-co-N,N'-bis(2-octyldodecyl)-naphthalene-1,4,5,8-bis(dicarboximide) derivative with extended glycolation (P-100)¹³ and the aforementioned NDI homopolymer p(gNDI-gT2).¹⁴ Using the P-90 best-performing concentration (40 mol%) as a guide, we found that TBAF indeed increases g_m in both polymers. The results are shown in **Figure S11**. In particular, P-100 $g_{m \max}$ increases by a remarkable 9x from P-100:TBAF(0 mol%) to P-100:TBAF(40 mol%), when taking the comparative film thicknesses into account. The thickness-normalised P-100:TBAF(40 mol%) $g_{m \max}$ is ca. 1.5 S/cm, which is higher than all NDIs reported to-date. We also note that I_{OFF} increases and V_T shifts towards more negative voltages, as expected for molecularly n-doped transistors. Although the impact of TBAF is less in p(gNDI-gT2), we still find that $g_{m \max}$ doubles, I_{OFF} increases and V_T shifts to more negative voltages with the addition of TBAF (**Figure S11d-e**); however, substantial OECT device variation inhibits statistical analysis, whilst suggesting that further optimization of the doping concentration is required to harness the intrinsic potential of n-doped p(gNDI-gT2).”

Reviewers' Comments:

Reviewer #1:

Remarks to the Author:

I had already very much liked the manuscript in the first round of revisions and suggested to publish it with minor corrections. The authors have now responded to all my comments in a convincing and detailed way, making the conclusions better substantiated. They also respond convincingly to the two other Referees concerning novelty. For me, the manuscript is now ready for publication.

Reviewer #2:

Remarks to the Author:

In this version the authors addressed all the reviewer's comment and added some relevant data. However I still believe the this manuscript lack in novelty and impact for the standards of Nature Communications. Can the authors explain more why they system(s) would impact significantly the bioelectronics community?

Reviewer #3:

Remarks to the Author:

The authors have answered my questions. It has been demonstrated that the technique used in this work is an effective strategy to improve the stability and transconductance of OECT. And it has also been shown that this method has a relatively good universality. However, the device performance such as transconductance, stability and mobility is not significantly improved compared with previously published works about OECT. Thus, I still think this manuscript is not suitable for the publication in Nature Communications. In addition, the authors should check the unit of drain current in Fig S1.

We thank the reviewers for their helpful comments and taking the time to review our manuscript for a second time. Below are our responses which describe any further changes made in our revised manuscript.

Sincerely,
Sahika Inal and co-authors

REVIEWER #1

REVIEWER: I had already very much liked the manuscript in the first round of revisions and suggested to publish it with minor corrections. The authors have now responded to all my comments in a convincing and detailed way, making the conclusions better substantiated. They also respond convincingly to the two other Referees concerning novelty. For me, the manuscript is now ready for publication.

AUTHOR'S REPLY: We thank the reviewer for their positive comments, as well as their previous suggestions and question that gave us the opportunity to strengthen our paper and the key messages.

REVIEWER #2

REVIEWER: In this version the authors addressed all the reviewer's comment and added some relevant data. However I still believe the this manuscript lack in novelty and impact for the standards of Nature Communications. Can the authors explain more why they system(s) would impact significantly the bioelectronics community?

AUTHOR'S REPLY: We thank the reviewer for considering the revised version of our manuscript. We have now added, on *page 23 and 24*, an outlook on the potential impact of water-stable n-doped organic electrochemical transistor demonstration on future bioelectronics research and technology in the Discussion. As for novelty, shortly, our work is the first-time demonstration of n-doping in an n-type OECT which not only enhanced the steady state performance but also resulted in devices with high operational stability when operated for long periods of time in biologically relevant media and long shelf life when left in buffer. The latter is particularly important considering the well-known but not yet resolved issues on the stability of n-type semiconductors and a challenge to overcome for the use of this new materials class in bioelectronics.

REVIEWER #3

REVIEWER: The authors have answered my questions. It has been demonstrated that the technique used in this work is an effective strategy to improve the stability and transconductance of OEET. And it has also been shown that this method has a relatively good universality. However, the device performance such as transconductance, stability and mobility is not significantly improved compared with previously published works about OEET. Thus, I still think this manuscript is not suitable for the publication in Nature Communications. In addition, the authors should check the unit of drain current in Fig S1.

AUTHOR'S REPLY: We thank the reviewer for their positive comments on the key messages from our paper and that doping is an effective and universal strategy; we believe this is the important key message. We have now included a new table, as *Supplementary Table 1*, to give a detailed comparison of the relevant organic electrochemical transistor performance metrics reported in this work and the published literature (note that only 4 n-type polymers have shown sufficient performance in electrolyte gated OEETs). While different geometries, batches and device operation conditions make such comparisons difficult, we included geometry normalized values, which altogether summarize once more the dramatic improvements provided by n-dopants. While it has been a synthetic challenge to develop n-type polymers with moderate stability and capacitance at the electrolyte interface, this work sets the stage for the use of n-doping as a strategy to bring the performance of n-type mixed conductors closer to that of their p-type counterparts.

We also thank the reviewer for highlighting the typographical error in Figure S1; indeed, all of the units should be in μA and we have now corrected this.